# Astrocytes enhance plasticity response during reversal learning
Lorenzo Squadrani [1], Carlos Wert-Carvajal[1], Daniel Müller-Komorowska[2], Kirsten Bohmbach [3], Christian Henneberger [3,4], Pietro Verzelli[1,5] & Tatjana Tchumatchenko [1,5]

Astrocytes play a key role in the regulation of synaptic strength and are thought to orchestrate synaptic plasticity and memory. Yet, how specifically astrocytes and their neuroactive transmitters control learning and memory is currently an open question. Recent experiments have uncovered an astrocyte-mediated feedback loop in CA1 pyramidal neurons which is started by the release of endocannabinoids by active neurons and closed by astrocytic regulation of the D-serine levels at the dendrites. D-serine is a co-agonist for the NMDA receptor regulating the strength and direction of synaptic plasticity. Activity-dependent D-serine release mediated by astrocytes is therefore a candidate for mediating between long-term synaptic depression (LTD) and potentiation (LTP) during learning. Here, we show that the mathematical description of this mechanism leads to a biophysical model of synaptic plasticity consistent with the phenomenological model known as the BCM model. The resulting mathematical framework can explain the learning deficit observed in mice upon disruption of the D-serine regulatory mechanism. It shows that D-serine enhances plasticity during reversal learning, ensuring fast responses to changes in the external environment. The model provides new testable predictions about the learning process, driving our understanding of the functional role of neuron-glia interaction in learning.

Synaptic plasticity, the modification of neuronal connections over time, is a fundamental brain process underlying learning and memory. Since its conceptualization, the study of synaptic plasticity has primarily focused on neurons' activity and interactions[1]. This neuron-centric view has been instrumental in understanding the basic mechanisms underlying synaptic changes. However, a growing body of experiments has started to unravel the critical role of glial cells, particularly astrocytes, in learning[2-4]. Traditionally viewed as mere support cells, astrocytes are now recognized as key players in a variety of neuronal functions, including synaptic transmission, synaptic plasticity, and memory formation[5-10]. See also the recent review by Linne[11]. Despite this emerging importance, a significant gap persists in our understanding of the precise mechanisms through which astrocyte-neuron interactions influence synaptic plasticity. This gap represents a critical area for exploration, as bridging it could significantly enhance our understanding of the cellular and molecular underpinnings of learning processes.

The field of synaptic plasticity has greatly benefited from the development of theoretical models. These models serve as essential tools for synthesizing experimental data, generating hypotheses, and guiding research.

Among them, phenomenological models like the BCM (Bienenstock, Cooper, and Munro) theory stand out[12,13]. Built on a concise set of assumptions, this theory accurately describes the development of pattern selectivity in cortical neurons[14,15] and predicted phenomena that were yet to be observed[16-18]. However, while phenomenological models are of great use in studying the brain's functions, they provide a limited understanding of the underlying biological mechanisms[19,20]. To fill this gap, it is necessary to develop detailed biophysical models capable of explaining the phenomenology.

The core postulate of the BCM theory is the existence of a threshold for long-term potentiation (LTP) which is dynamically regulated by the history of post-synaptic activity. Several hypotheses were proposed in the last decades to give a physiological justification for this postulate[18,21-24]. Many of them involve intracellular calcium dynamics in the post-synaptic neuron. However, none of the hypotheses have found a clear experimental confirmation yet, and the underlying mechanism remains a matter of debate[13,25]. At the same time, recent experiments point at the extracellular regulation of D-serine, mediated by astrocytes, as a previously overlooked candidate for connecting BCM theory with the intricate biology of the brain.

[1]Institute of Experimental Epileptology and Cognition Research, Medical Faculty, University of Bonn, Bonn, Germany. [2]Okinawa Institute of Science and Technology Graduate University, Okinawa, Japan. [3]Institute of Cellular Neurosciences, Medical Faculty, University of Bonn, Bonn, Germany. [4]German Center for Neurodegenerative Diseases (DZNE), Bonn, Germany. [5]These authors contributed equally: Pietro Verzelli, Tatjana Tchumatchenko. ✉e-mail: pietro.verzelli@uni-bonn.de; tatjana.tchumatchenko@uni-bonn.de

D-serine is a co-agonist for the N-methyl-D-aspartate receptor (NMDAR)[26,27], and its synaptic levels determine NMDAR tone and NMDA-dependent plasticity[6,28–30]. D-serine was shown to affect the activity-dependence of long-term synaptic changes, in a manner similar to the shifting of a potentiation threshold, resembling that of the BCM theory[31]. More recently[32], described a new mechanism for the astrocytic regulation of synaptic available D-serine, characterized by a bell-shaped dependence on the frequency of the post-synaptic activity. The mechanism exhibits remarkable similarities to the core postulates of BCM theory and is shown to impact behavioral choices in mice when it is disrupted.

In this work, we tackle the following research questions: (1) what is the functional role of astrocyte-mediated regulation of D-serine gliotransmitter in learning? (2) What are the biophysical mechanisms underlying BCM plasticity and its phenomenology? In particular, we explore the hypothesis that the astrocytic regulation of D-serine represents the physiological implementation of the BCM dynamic threshold; we refer to this as the "D-serine hypothesis". To support it, we present a formal derivation of the BCM rule, starting from the mathematical description of the astrocytic mechanism observed in ref. 32. Building upon the hypothesis, we develop a mathematical model capable of explaining precise behavioral effects resulting from astrocytic manipulation, as observed in experiments, and produce new testable predictions on the learning process and the functional role of D-serine in the brain. Overall, our results contribute to reducing the gap between our theoretical understanding of plasticity and the experimental knowledge of neuron-glia interactions in learning.

## Results

### The D-serine hypothesis: astrocytes orchestrate BCM plasticity

Recent experiments on mice CA1 hippocampal neurons have shed light on a novel molecular mechanism, forming a feedback loop through which the activity of pyramidal neurons influences their future dynamics and plasticity[32]. The feedback loop is mediated by astrocytes and is enacted by two main molecules: endocannabinoids released by neurons upon activation, which bind to the cannabinoid receptors (CBRs) of astrocytes[33,34], and D-serine, released into the extracellular environment in response to astrocytic calcium signaling triggered by CBRs activation[6,28,35,36] (Fig. 1a).

Let $v$ represent the average firing rate of a neuron population, and $d$ denote the concentration of D-serine in the extracellular environment. Based on previous observations, we describe the dynamics of $d$ so that it follows a given function of the post-synaptic activity, denoted as $D(v)$, with a time constant $\tau_d$:

$$\dot{d} = -\frac{1}{\tau_d}(d - D(v)) \qquad (1)$$

The right-hand side of Eq. (1) can be interpreted as the sum of a D-serine degradation/uptake term and a term that describes activity-dependent D-serine supply. The function $D(v)$ describes how post-synaptic activity influences D-serine concentration levels. The dependence of the D-serine release on the post-synaptic activity has been studied by monitoring D-serine-dependent dendritic integration during the axonal

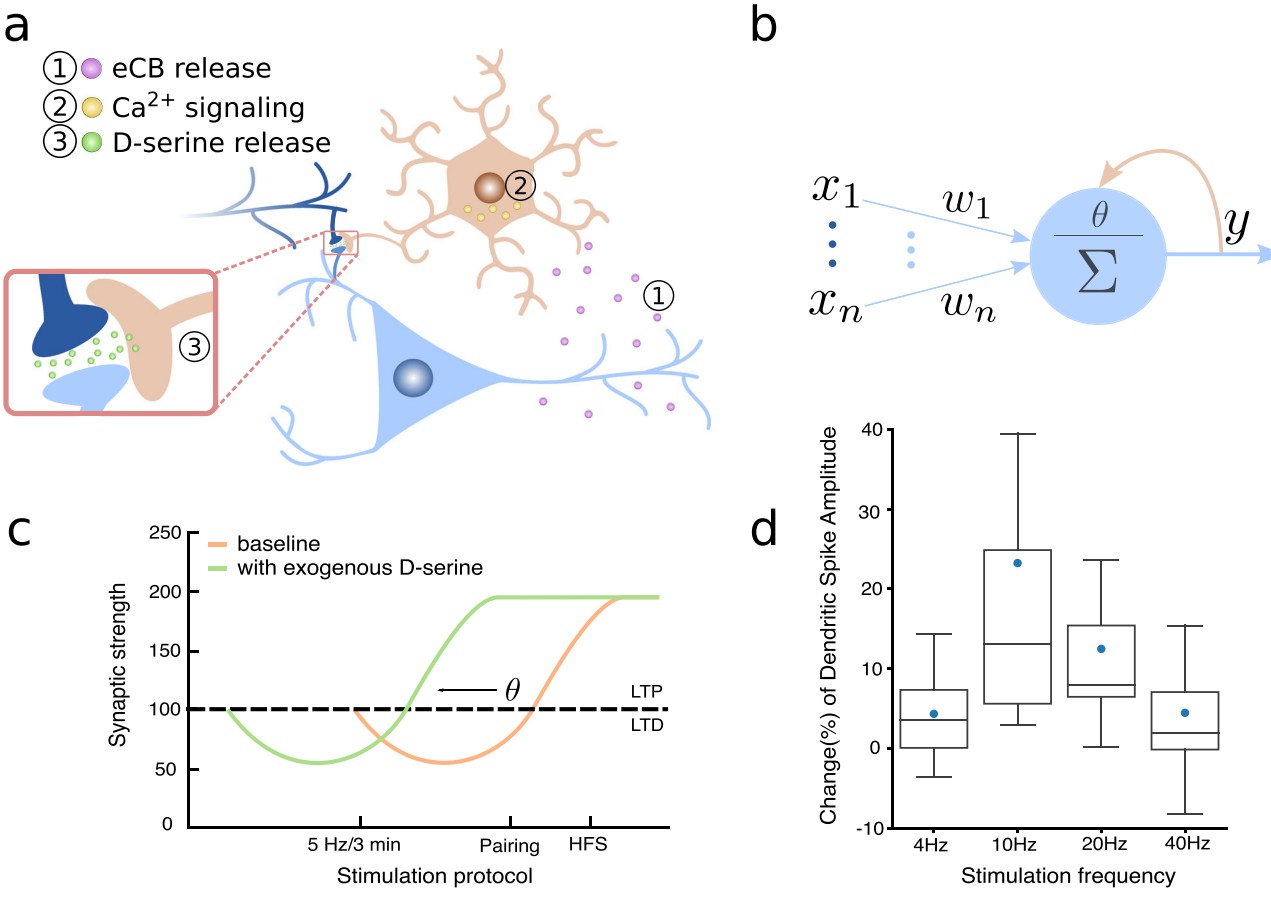

**Fig. 1 | The D-serine hypothesis. a, b** The astrocytes-mediated feedback loop observed in ref. 32 shows remarkable similarities with the BCM postulate of a synaptic modification threshold $\theta$ sliding as a function of the post-synaptic activity $y$. The feedback loop is described in three steps: (1) endocannabinoids are released by the post-synaptic neuron upon activation, (2) they bind with astrocytic cannabinoid receptors and trigger an intracellular calcium signaling, which results in (3) the release of D-serine at the neuron dendrites. **c** The increase (decrease) of D-serine concentration at the synapse promotes LTP (LTD), which can be interpreted as the shifting of a threshold for LTP, like the BCM one (figure adapted from ref. 31). **d** The astrocytic regulation of D-serine presents a peculiar bell-shaped dependence on the frequency of post-synaptic stimulation, analogous to the super-linear dependence of the BCM's threshold on the post-synaptic activity. Changes in the D-serine level are indirectly measured through changes in the amplitude of dendritic spikes' slow component (figure adapted from ref. 32).

stimulation of CA1 pyramidal neurons at different frequencies[32]. Notably, the D-serine concentration level exhibits a bell-shaped dependence on the frequency of stimulation, with a peak at 10 Hz (Fig. 1d). Based on this, we choose the simple mathematical form:

$$D(\nu) = D_0 - a(\nu - \nu_0)^2 \tag{2}$$

where $D_0$ is the maximum level of D-serine concentration, occurring when $\nu = \nu_0$, and $a$ is a constant with the appropriate dimensions ($[a] = [D] \cdot [\nu]^{-2} = [N] \cdot [L]^{-3} \cdot [T]^2$, where $N$, $L$ and $T$ are the amount of substance, length, and time dimensions respectively).

The importance of D-serine stems from its interaction with synaptic NMDARs, which play a key role in the induction of synaptic plasticity[37]. D-serine is an essential endogenous ligand for the co-agonist site of synaptic NMDARs in the brain[26,27,31]. Consequently, its presence is required for the activation of NMDARs. Consistent with these findings, alterations in D-serine levels have been unequivocally associated with changes in the properties of activity-dependent long-term plasticity[31]. In particular, the same stimulation protocol induced LTP or longterm depression (LTD) depending on the level of D-serine. In the presence of a high (low) level of D-serine, LTP (LTD) is facilitated, and the same effect is equally seen with three different stimulation protocols. Such observation suggests that the effect of D-serine alterations can be interpreted as a shifting of a threshold for LTP induction (Fig. 1c). We denote with **w** a vector of synaptic strengths and with **x** a vector of pre-synaptic activities. We define the integration of the input vector with the synaptic strengths to track the neural activity with respect to the reference value $\nu_0$, thus $y = \nu - \nu_0 = \mathbf{w} \cdot \mathbf{x}$. The operating regime of our model will be above $\nu_0$, $y > 0$ because this is best supported by the available experimental data. Let us mention that firing rates below this regime could be described by other plasticity mechanisms that are outside of our model framework. Finally, we define an update rule for the synaptic strengths in such a way that the direction of plasticity depends on the level of D-serine through a function $\theta(d)$:

$$\tau_w \dot{\mathbf{w}} = \mathbf{x}y(y - \theta(d)) \tag{3}$$

where $\tau_w$ is a time constant. The proportionality of the update to the pre-and post-synaptic firing realizes the Hebbian principle, while the quadratic dependence on the post-synaptic rate has been suggested by experimental measurements (Ref. 16,31) and has been derived from spikes-based learning rules (STDP[38,39], calcium-based plasticity[40]). The equation is formally equivalent to the BCM update rule[12], with the important difference that here the threshold is no more a postulated dynamical variable but a function of the D-serine concentration. We show now that, choosing the function $\theta(d)$ according to experimental observations and considering the dynamics of D-serine described by Eq.s (1) and (2), we obtain a dynamical system equivalent to BCM.

According to the fact that a higher level of D-serine facilitates LTP, we choose the threshold function to be linearly decreasing with respect to the D-serine concentration:

$$\theta(d) = b(D_0 - d) \tag{4}$$

where $b$ is a constant with the appropriate dimensions ($[b] = [y] \cdot [d]^{-1} = [T]^{-1} \cdot [N]^{-1} \cdot [L]^3$). The threshold is 0 when $d$ is at its maximum value $D_0$, and it increases as $d$ decreases. We define the new variable $\theta = b(D_0 - d)$, and substitute it in Eq.s (1) and (3), obtaining:

$$\begin{aligned} \tau_w \dot{\mathbf{w}} &= \mathbf{x}y(y - \theta) \\ \tau_d \dot{\theta} &= -\theta + ab \cdot y^2 \\ y &= \mathbf{w} \cdot \mathbf{x} \end{aligned} \tag{5}$$

which is equivalent to the BCM plasticity rule[12]. To minimize the number of free parameters and be as consistent as possible with respect to

previous parameter settings in the BCM model we set $ab = 1$ for all our subsequent model simulations. However, during the model formulation stage, we thought it could be helpful to think of $a$ as a proportionality factor between firing rate and D-serine amount and of $b$ as a proportionality factor mediating the transformation between threshold and D-serine, since both could be altered individually in future studies or modified experimentally.

## Testing the D-serine hypothesis within a behavioral context

The biological interpretation of BCM proposed above allows us to explore how the disruption of the D-serine feedback loop changes learning performance. In mice, this intervention has been shown to affect learning performance during a place avoidance task[32]. The original experiment (Fig. 2) consisted of a mouse freely exploring an O-shaped maze, triggering an air puff each time it passed through a fixed position; after the mouse learned to avoid that position, the puff was moved to the opposite side. Learning was assessed by measuring the number of visits to each position. The disruption of the D-serine feedback loop corresponded to a deletion of astrocytic type 1 CBRs in a group of mice (mutant mice). In the initial phase of the task, mutant mice learned to avoid the air puff correctly, en par with the control mice (Fig. 2b). However, when the puff position was switched, the mutant showed *slower learning* with respect to control mice. The mutation thus has a significant and specific impact on behavior, which cannot be described as a generic learning impairment, offering an ideal test bench for our hypothesis.

To approach this problem within the theoretical learning framework described in section, we model a simplified version of the task with a two-state environment, $S^1$ and $S^2$ (Fig. 3). Such a description is suitable because the only relevant measurements in the original experiment were the number of visits to the puff position. The decision process of the mouse is described by a single action unit (Fig. 3, bottom). Each state $S^i$ is associated with a one-hot encoding vector $\mathbf{x}^i$ serving as input for the action unit. The agent-mouse decides to switch state with a probability directly proportional to the activation $y(\mathbf{x}^i)$ of the action unit. For a detailed description of how the agent moves between the states refer to the section *Methods*. The simulation is divided into two phases. At the beginning, a punishing signal is associated with $S^2$ (Phase 1); the agent has to learn to avoid the punishing state. After a given amount of time has elapsed, the punishing signal is moved to $S^1$ (Phase 2), so that the agent needs to learn the new configuration, reverting the previously learned behavior. Learning emerges from the adaptation of the synaptic weights of the action unit, which is implemented with the BCM plasticity model, and modified in order to account for the punishing signal.

## Extending BCM to solve a reinforcement-driven task

The original BCM model, as presented by Bienenstock, Cooper, and Munro[12], is a purely unsupervised learning rule. Consequently, it is insufficient for behavioral learning tasks that require distinguishing between neural activities associated with positive or negative outcomes. It is widely acknowledged that such distinction is mediated in the brain through the actions of neuromodulators, such as dopamine[41,42]. In the realm of theoretical modeling, neuromodulation has been described by extending two-factor learning rules (where the factors are pre- and post-synaptic activities) to *three-factor* learning rules, with the third factor being a reinforcement-signaling variable[43]. While this problem has been addressed for general Hebbian learning and Spike Timing Dependent Plasticity[44–47], it has not been explored for BCM. Here, we propose a tractable mathematical model that allows for studying the punishment-driven place avoidance task described in the previous section.

To this aim, we first rewrite the system (5) for discrete time steps, in consistency with the literature on reinforcement learning[20]; for each transition from state $S_t$ to state $S_{t+1}$, the unit with an activity $y_t$ determined by the weights $\mathbf{w}_t$ and the input $\mathbf{x}_t = \mathbf{x}(S_t)$, the BCM update rule reads:

$$\Delta\mathbf{w}_t = \frac{1}{\tau_w}\mathbf{x}_t y_t (y_t - \theta_t) \tag{6a}$$

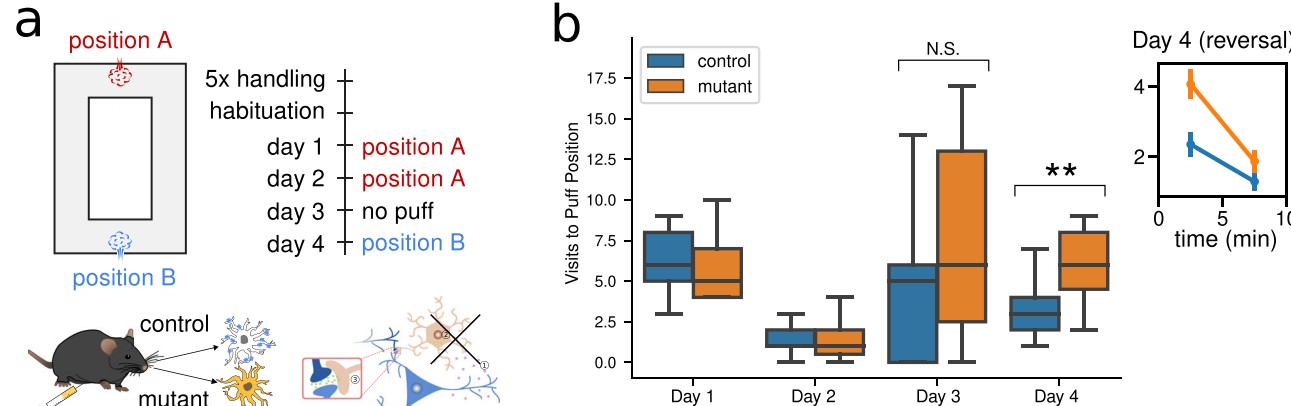

**Fig. 2 | Learning impairment of mutant mice during a place-avoidance. a** The learning ability of the mouse was experimentally tested with a place-avoidance task driven by an aversive stimulus. The mouse was free to explore an O-shaped maze, triggering an air puff each time it passed through a fixed location; after the mouse learned to avoid that position, the puff was moved to another location. The test is repeated with two groups of mice: a control group, and a group in which the astrocytic type 1 cannabinoid receptors had been knocked out (mutant). **b** Mutant mice showed slower learning with respect to control mice, only during reverse learning (Day 4). On each day, the mouse is free to explore the maze for a single session of 10 minutes, and the number of visits to the puff position is recorded. The

box plots represent the statistics from 15 (control) and 17 (mutant) animals. There is no significant difference between the groups on Day 1, 2, and 3 (Day 1: 6.59 ± 0.73 vs. 6.13 ± 0.62, $p = 0.62$; Day 2: 1.41 ± 0.21 vs. 1.40 ± 0.34, $p = 0.73$; Day 3: 6.00 ± 1.50 vs. 7.27 ± 1.45, $p = 0.45$; two-sided Mann-Whitney U-test), while there is a very significant difference on Day 4 (Day 4: 3.65 ± 0.50 vs. 5.93 ± 0.61, $p = 0.0066$, two-sided Student's $t$-test). The inset axis shows the same data for Day 4, binned into two 5-minute intervals. Each point is computed as the average over all the animals, and the error bar is the standard error. In the first 5 minutes, the mutant mice visited the puff position significantly more than the control mice, showing a learning deficit. In the last five minutes, the difference is no more observed. Figures adapted from ref. 32.

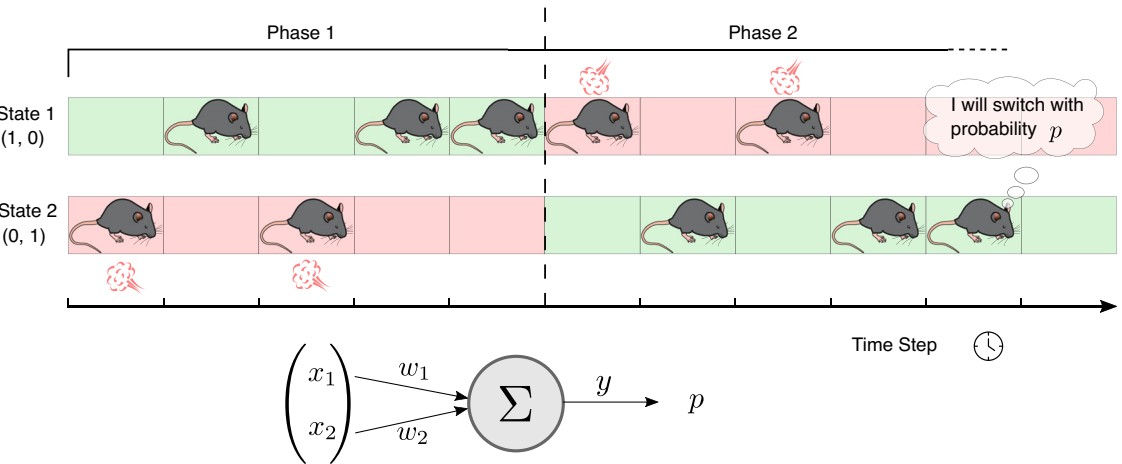

**Fig. 3 | Simulation of the passive place-avoidance task.** In our simulation of the experiment, we simplify the movement of the mouse as a transition between two locations (State 1 and State 2 in the figure), one of which is associated with a punishment (red background). The simulation is divided into two phases. During the first phase, State 2 is punishing, while in the second phase, the punishment is moved to State 1. At each time step, the agent mouse decides whether to stay in the

current state or to switch. The decision is based on the current state, represented by a one-hot encoding state vector, and on the information the agent-mouse has about the states, encoded in the weights $w_1$ and $w_2$ of a single action unit. The integration of the state vector with the synaptic weights determines the activity $y$ of the unit, which in turn determines the probability $p$ of switching state.

$$\Delta\theta_t = -\frac{1}{\tau_\theta}\left(\theta_t - y_t^2\right). \quad (6b)$$

Following the ideas discussed above, we introduce a third, reinforcement factor in Eq. (6a):

$$\Delta w_t = -R(S_{t+1})\mathbf{x}_t y_t(y_t - \theta_t). \quad (7)$$

Note that the value of the reinforcement factor $-R(S_{t+1})$ depends on the destination state, providing information about the outcome of the action which was taken at time $t$[43,45].

We refer to the resulting plasticity rule as 'Reinforcement-BCM' (R-BCM). From a functional perspective, BCM plasticity allows the action unit

to develop selectivity with respect to the incoming stimuli, thus making the agent select one of the two states. The extension to the R-BCM allows the agent to select one specific state according to the values of the reinforcement factor, and to change selection when the values are switched. Thus, the new rule can solve the behavioral task described in the previous section. Further details and the analysis of stability are provided in *Methods*.

**Simulations reproduce learning deficit in mutant mice**

The learning deficit observed in ref. 32 following the disturbance of neuron-astrocyte communication exhibits distinct characteristics, which present challenges in directly mapping it to astrocytic and neuronal activity. Naively incorporating the effect of the disruption of D-serine regulation as a generic learning impairment (for example by linking it to a larger time scale of the weight dynamics) would not align with the experimental findings, as mutant

**Fig. 4 | The model reproduces learning deficit in mutant mice.** The figure shows the results of the simulated place-avoidance experiment (described in Fig. 3) for control mice and mutant mice. The top graph shows the rate of visit to the state $S^1$, computed as the relative number of visits to state $S^1$ in the last 1000 time steps. The bottom graph shows the evolution of the synaptic weights. All the curves are averages over 50 simulated mice. During the first phase of the simulation (left of the black line) the states $S^1$ and $S^2$ are associated with a positive and negative reinforcement signal, respectively. Coherently, the frequency of visits of the agent-mouse to state $S^1$ increases to about 9 times out of 10. Both weights converge to a finite value, one of them close to zero, the other close to 0.4. There is no difference in learning between control mice and mutant mice. During the second phase (right of the black line), the reinforcement signals are switched. Both groups of mice learn to avoid the state $S^1$, with a visit frequency of about 1 time out of 10. However, mutant mice show a significant delay during this phase of reverse learning. During this phase, the weights switch their values. The switching is faster in control mice (top) than in mutant mice (bottom). More precisely, the higher weight decays with the same speed in both groups, reflecting the initial drop in the visiting frequency (top figure, second phase). However, the low weight increases earlier and faster in control mice than in mutant mice.

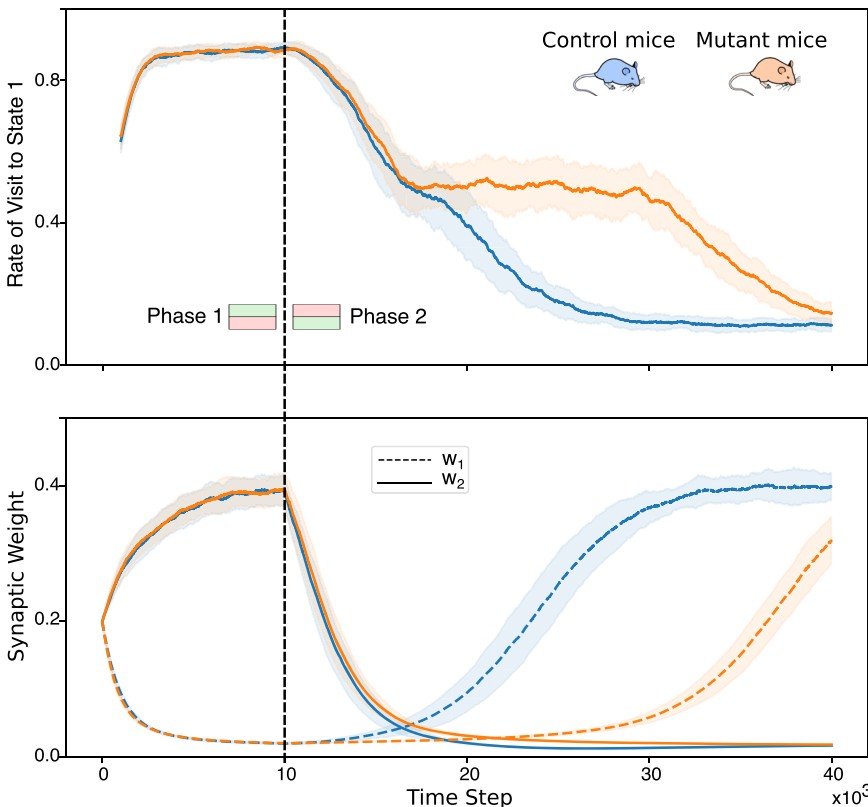

mice exhibited a deficit only in the second phase of learning, when the puff position is switched. Importantly, our biological interpretation of BCM allows us to give a more precise description of the disruption of the D-serine feedback loop, by modifying the dynamics of D-serine. Experiments show that, after the deletion of astrocytic type 1 CBRs, the D-serine levels are not affected anymore by neuronal stimulation. The concentration of D-serine is thus constant (non-zero because this would completely prevent NMDA-dependent plasticity). This aspect is described by Eq.(2) by setting a zero value of the activity $D(0) = D_0 - a\nu_0^2$. In practice, this corresponds to a constant value for the LTP threshold $\theta = \theta_0$.

The mice moving in the two-state environment and utilizing R-BCM for synaptic weight updates learn to avoid the state associated with the negative reinforcement (Fig. 4, top). In the initial phase of the simulation, states $S^1$ and $S^2$ are linked to positive and negative reinforcement signals, respectively. Consequently, the frequency of state visits $S^1$ increases, occurring approximately 9 out of 10 times. The same behavioral pattern occurs when the LTP threshold is fixed, simulating the disruption of the D-serine feedback loop in mutant mice. This behavior aligns with the experimental findings in ref. 32 since both control and mutant mice developed place-avoidance behavior.

The frequency at which each state is visited is determined by the synaptic weights of the action unit, as described in the *Methods* section. We plot the weights' evolution throughout learning in Fig. 4, bottom graph. During Phase 1, the weight $w_1$, which represents the probability of switching state from $S^1$ to $S^2$ (the punishing state), correctly converges to a value close to zero. The weight $w_2$, which represents the probability of switching state from $S^2$ to $S^1$, converges to a higher value, but lower than 0.5. This means that the probability of leaving the punishing state is not close to 1, as might be intuitively expected from a probabilistic agent who learned to avoid this situation. However, this property can be justified and interpreted as proof of the coherence of the model. The *avoidance* behavior can be conceptually divided into two sub-behaviors: *going away from* the aversive situation and *not going back to* it. Such a distinction is important because the two behaviors are unlikely to arise from the same neurobiological processes. For

instance, the *not going back to* behavior requires memory of past experiences, while the *going away from* does not since the aversive stimulus is present and can drive the behavior directly. Coherently, because we are only modeling one mechanism (synaptic plasticity), the model captures the *not going back to* behavior ($P(\text{stay}|\text{no punishment}) \approx 1$) but not the *going away from* ($P(\text{change}|\text{punishment}) \approx 0.5$).

After switching the reinforcement signals associated with the states (second phase of the simulation), the control mice quickly adapt their behavior to avoid the other state. In contrast, the mutant mice learn significantly slower (Fig. 4, top). Strikingly, such difference in the two phases of learning is again in agreement with what has been observed in ref. 32 (Fig. 2).

An inspection of the evolution of the weights reveals interesting insights into the deficit (Fig. 4, bottom). The weight associated with the previously punishing state drops down without significant difference in both classes of mice. However, the weight associated with the newly punishing state rises much slower in the mutant mice. The reason for that lies in the blockade of the D-serine regular dynamics, as seen by looking at the D-serine levels during reverse learning (Fig. 5, left). In control mice, reverse learning is accelerated by a temporary increase in the D-serine levels, which results in a lower LTP threshold.

Next, we tested whether increasing the intensity of the punishing signal (by modulating the values $R_i$, see the section *Methods*) for the mutant mice could restore the learning speed. In Fig. 5, right side, we plot the rate of the visit to state $S^1$ during the second phase of learning for control mice with $R_1 = -1$, and for mutant mice with $R_1 = -1, -1.2, -1.5$. The mutant mice learn faster as the magnitude of the $R_1$ increases, and it reaches the speed of the healthy mice for $R_1 = -1.5$.

## Discussion

We presented a mathematical model of synaptic plasticity incorporating a molecular interaction mechanism between astrocytes and CA1 pyramidal neurons. The mechanism, recently described in ref. 32, consists of a feedback loop that is initiated by the release of endocannabinoids by post-synaptic neurons and terminating with astrocytic regulation of synaptic D-serine

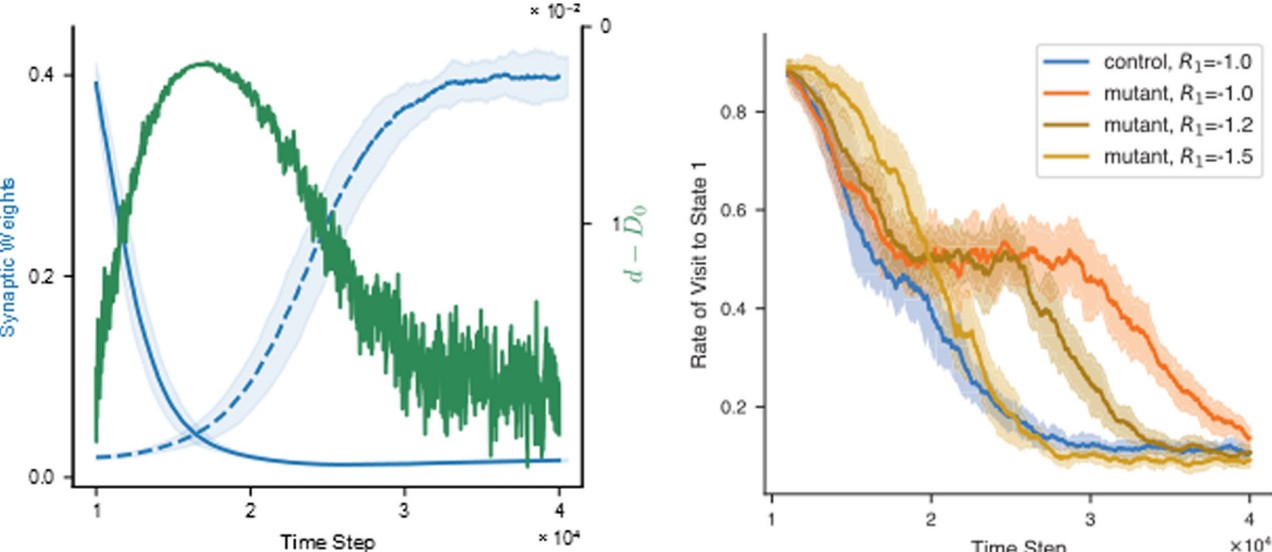

**Fig. 5 | (left) Increased D-serine levels during reverse learning.** The blue lines are the evolution of the two synaptic weights during reverse learning of normal mice (same lines of Fig. 4, bottom graph, second phase). The D-serine concentration relative to its maximum value $D_0$ is superposed (green line). The scale for $d - D_0$ is of course different from that of the synaptic weights, and it is drawn on the right axis. The D-serine levels increase during reverse learning and they return to the basal value when the switching is completed. This prediction suggests that the astrocytic regulation of D-serine has the specific function of enhancing plasticity response to changes in the external

environment (possibly via increasing the opening probability of NMDAR channels). (right) Increasing the aversiveness of the stimulus recovers learning speed for mutant mice. The rate of visit to state $S^1$ during the second phase of learning is plotted, for control mice with $R_1 = -1$, and for mutant mice with $R_1 = -1, -1.2, -1.5$. The mutant mice show a significant delay with respect to control, for the same values of the reinforcement variable (already discussed in Fig. 4). However, the mutant mice learn faster as the magnitude of $R_1$ increases, and it reaches the speed of the control mice for $R_1 = -1.5$.

levels. We showed that the firing rate-dependence of the feedback loop (Fig.1c, d), can be translated into a plasticity rule equivalent to the well-known Bienstock-Cooper-Munro learning rule. Bohmbach et al.[32] investigated the functional role of the feedback loop by testing how a mouse learns to avoid an aversive stimulus and then changing the position of the stimulus and observing the re-learning dynamics. The experiment showed that mice in which the astrocyte-neuron interaction is genetically disrupted present a specific learning deficit: they learn to avoid the aversive stimulus just like control mice, but they are significantly slower to reverse learning and learn anew after the position of the stimulus is changed. To test the ability of our model to capture the experimentally reported learning curves, we simulated an analogous place-avoidance task with a learning phase and a relearning phase. Our biophysical form of BCM allowed us to quantify the disruption of the astrocytic feedback loop. We found that our model simulations correctly reproduced the specific learning deficit observed in the experiments with the mutant mice showing a slow-down only during the reverse phase of learning Fig. 4.

The most important contributions of this work can be summarized in three points. First, we proposed a novel hypothesis for the molecular mechanisms underlying BCM-like plasticity in the brain. The BCM theory received multiple experimental confirmations and predicted phenomena that were yet to be observed[13]. However, being a phenomenological theory, it provides a limited understanding of the biological mechanisms underlying learning. Our work contributes to the development of a detailed biophysical model explaining BCM's phenomenology in CA1. Second, our model proposes a theoretical description of the astrocyte-neuron interactions to synaptic plasticity. Specifically, it points to a specific functional role for the astrocytic regulation of synaptic D-serine levels, i.e. ensuring a fast plasticity response to changes in the external environment (Fig. 5, left). The involvement of D-serine in synaptic plasticity and its possible relation to the BCM rule was already suggested in experimental literature[31,48], but it has been previously considered in theoretical frameworks. Finally, our model generated a number of testable predictions which can guide future experiments. For example, the learning curves depicted in Fig. 5 predict that an increase in

the extracellular D-serine level should be observed during the reverse learning phase. If confirmed by future experiments, this result points to an interesting functional role for the D-serine regulatory system, i.e. ensuring a fast plasticity response to changes in the external environment. Also the evolution of the synaptic weights during reversal learning in the mutant mice (Fig. 4, bottom) suggests that the learning impairment is due to the inhibition of LTP caused by astrocytic CBR knockout. The inhibition is not homogeneous across the range of synaptic strength however: in the first phase of learning, when the synaptic strength starts from an intermediate value, control mice and mutant mice do not show any difference in the LTP; in the second phase, when synaptic strength starts from a highly depressed value, LTP is slower in mutant mice compared with control mice.

Historically, the BCM model was designed to explain the development of orientation selectivity in cortical pyramidal neurons. Our derivation of the BCM rule, on the other hand, is based on experimental manipulations of astrocytes and D-serine dynamics in the CA1 region of the hippocampus, indicating that the establishment and perturbations of orientation selectivity and learning and re-relearning of aversive cues could share mechanistic similarities.

**Integrating diverse plasticity models: exploring extracellular and intracellular dynamics**

The D-serine hypothesis for the adaption of the BCM threshold considers the extracellular dynamics at the synapse, while the previously studied BCM mechanisms were built on the intracellular dynamics, usually of calcium[22], but also of other molecules like the kinase CamKII[24,49]. For this reason, our hypothesis does not directly contradict the previous mechanisms but illustrates that similar net effects can arise on extra and intracellular scales. The interaction between the extracellular synaptic dynamics and the intracellular dynamics which is mediated by the NMDARs dynamics, could be considered in future studies and provide a more general model of synaptic plasticity and extending the mathematical framework proposed in ref. 22.

From the computational point of view, the BCM theory has been related to Spike-Timing Dependent Plasticity (STDP) and it has been shown

## Table 1 | R-BCM dynamics

| Transition | Reinforcement | Unit activation | $\Delta w$ |
|---|---|---|---|
| $S^1 \longrightarrow S^1$ | $R_1$ | $\mathbf{w} \cdot \mathbf{x}_1 = w_1$ | $\Delta w_1 = -R_1 w_1(w_1 - \theta)$<br>$\Delta w_2 = 0$ |
| $S^1 \longrightarrow S^2$ | $R_2$ | $\mathbf{w} \cdot \mathbf{x}_1 = w_1$ | $\Delta w_1 = -R_2 w_1(w_1 - \theta)$<br>$\Delta w_2 = 0$ |
| $S^2 \longrightarrow S^1$ | $R_1$ | $\mathbf{w} \cdot \mathbf{x}_2 = w_2$ | $\Delta w_1 = 0$<br>$\Delta w_2 = -R_1 w_2(w_2 - \theta)$ |
| $S^2 \longrightarrow S^2$ | $R_2$ | $\mathbf{w} \cdot \mathbf{x}_2 = w_2$ | $\Delta w_1 = 0$<br>$\Delta w_2 = -R_2 w_2(w_2 - \theta)$ |

At each step, the agent performs one of the state transitions in the first column of the table. The reinforcement signal is determined by the final state. The action unit activity is determined by the initial state. The synaptic weights are updated according to Eq. (7). We note that since $\Delta\mathbf{w}$ is proportional to the one-hot input vectors, only one of the weights changes at each transition.

that a BCM-like rate-based plasticity can arise from STDP through an averaging operation[38]. In this context, the threshold for synaptic potentiation depends on the shape of the STDP window. According to the D-serine hypothesis, the threshold is a function of the synaptic D-serine concentration. To understand the relation and the compatibility of the two models it is thus crucial to understand how different D-serine levels affect the STDP window. This question could be tackled computationally in future studies by modeling how NMDARs shape the STDP window. Interestingly, the authors in ref. [50] have shown that an STDP model in which the synaptic change window depends on the time of the last post-synaptic spike preceding the plasticity-inducing spikes pair (triplet STDP) can be linked to the BCM rule. This suggests that D-serine could be affecting the STDP windows and STDP learning rules involving multiple pre- and post-synaptic spike times.

### Exploring stability mechanisms in synaptic plasticity

The standard Hebbian learning rule is known to generate maximally strong synaptic weights and lead to network instability[51]. High neural activity leads to synaptic potentiation, which in turn amplifies future activity. This process is sometimes referred to as the Hebbian positive feedback mechanism. The BCM theory addresses the problem by introducing the sliding threshold mechanism[17,23], which can be implemented in the brain through the D-serine feedback loop. However, it has been observed that learning can still occur even when the D-serine feedback loop is disrupted, suggesting the existence of alternative forms of stabilization. The reinforcement-BCM model proposed in this work successfully reproduces the effects of D-serine manipulation on learning and offers an alternative source of stability through the zero-averaging of positive and negative reinforcement signals. However, this solution has some inherent limitations. It is not easily applicable in the absence of reinforcement or when only positive or negative reinforcement is present. Additionally, it provides stability on average, which restricts the learning speed as excessively rapid weight updates could compromise the system's averaging operation and lead to instability.

When the BCM mechanism of stabilization based on a sliding potentiation threshold was initially proposed, experimental research was conducted to investigate its biological plausibility[17,52]. The studies aimed to determine whether the recent history of synaptic activity could influence the induction of LTP. It was observed that the induction of LTP under a specific stimulation protocol was not fixed but could indeed be strongly influenced by the prior synaptic activity. However, the influence on LTP induction was found to be specific to the stimulated synapses rather than generalized to all inputs onto the postsynaptic neuron, as postulated by the BCM theory. A similar contradiction arises from our reformulation of the BCM model. We assumed that the D-serine level, represented by the variable $d$ in Eq. (1), is the same at all synapses, which led to a neuron-wide potentiation threshold identical to BCM. However, considering the close proximity of astrocytes to dendrites, it is possible that their regulatory action on D-serine can target

individual or specific groups of synapses, posing a contradiction to the neuron-wide potentiation threshold assumption in the BCM model. This possibility will be further explored when extending the D-serine hypothesis to a network of neurons, possibly providing new insights into the intricate interactions between neurons and glial cells.

## Methods

### Model of the place-avoidance task

We simulate a mouse performing a place avoidance test with an aversive stimulus. The spatial environment is modeled as a discrete two-state space $S^1, S^2$. At each time step, the agent chooses between two actions: staying in the current state or moving to the other one. The states are represented by one-hot encoding vectors $\mathbf{x}^1 = (1, 0)$ and $\mathbf{x}^2 = (0, 1)$. The vectors are used as synaptic inputs for a single action unit, describing the decision process of the agent-mouse. If at time step $t$ the agent-mouse occupies the state $S_t$ with associated input vector $\mathbf{x}_t$, the activation of the action unit is given by the scalar product of the input vector with its synaptic weights:

$$y_t = \mathbf{w}_t \cdot \mathbf{x}_t \qquad (8)$$

The initial values of the weights are $(w_1, w_2) = (0.2, 0.2)$ and the agent starts from a random state. The agent-mouse switches state at the next time step with a probability $p_{\text{change}}$ directly proportional to the firing rate of the action unit, with a minimum value of 0.05:

$$p_{\text{change}} = \max(0.05, y_t). \qquad (9)$$

The lower bound of $p_{change}$ was introduced to maintain a minimum level of exploration. After each transition, the synaptic weights and the synaptic threshold (i.e. the D-serine level) are updated:

$$\mathbf{w}_{t+1} = \mathbf{w}_t + \Delta\mathbf{w}_t$$
$$\theta_{t+1} = \theta_t + \Delta\theta_t$$

The learning rule consists of the BCM plasticity rule modified in order to account for the negative reinforcement (see section and the rest of the Methods).

### Learning rule

On each transition from state $S_t$ to state $S_{t+1}$, the unit having an activity $y_t$ determined by the weights $\mathbf{w}_t$ and the input $\mathbf{x}_t = \mathbf{x}(S_t)$, the weights are updated according to the following learning rule:

$$\tau_w \Delta\mathbf{w}_t = -R(S_{t+1})\mathbf{x}_t y_t(y_t - \theta_t)$$
$$\tau_\theta \Delta\theta_t = -\theta_t + y_t^2. \qquad (10)$$

This is the BCM learning rule with the addition of a multiplicative factor depending on the final state. The equation for the weight change is dissected into four possible cases (the four possible transitions) in Table 1. We refer to the rule as "R-BCM".

We compute the stationary points of R-BCM with the same procedure used for obtaining the stationary points of the original BCM[25,53], i.e. replacing the stochastic differential equation (7) with the same equation averaged over the input environment:

$$E[\Delta w] = -p_1\mathbf{x}_1(p_{11}R_1 + p_{21}R_2)y_1(y_1 - \theta)$$
$$-p_2\mathbf{x}_2(p_{11}R_1 + p_{21}R_2)y_1(y_1 - \theta) \qquad (11)$$

where $y_1 = \mathbf{w} \cdot \mathbf{x}_1$, $y_2 = \mathbf{w} \cdot \mathbf{x}_2$, $p_i$ is the probability of being in state $i$, and $p_{ij}$ is the probability of the transition from state $j$ to state $i$. The dynamics of the agent can be described as a Markov chain over the two-state space so that the probabilities $\mathbf{p} = (p_1, p_2)$ obey $\mathbf{p}(t+1) = \Pi\mathbf{p}(t)$, where $\Pi$ is the transition matrix. We recall that the probability of switching state is related to the firing rate of the action unit through Eq. (9). Thus, for a fixed weight vector, the

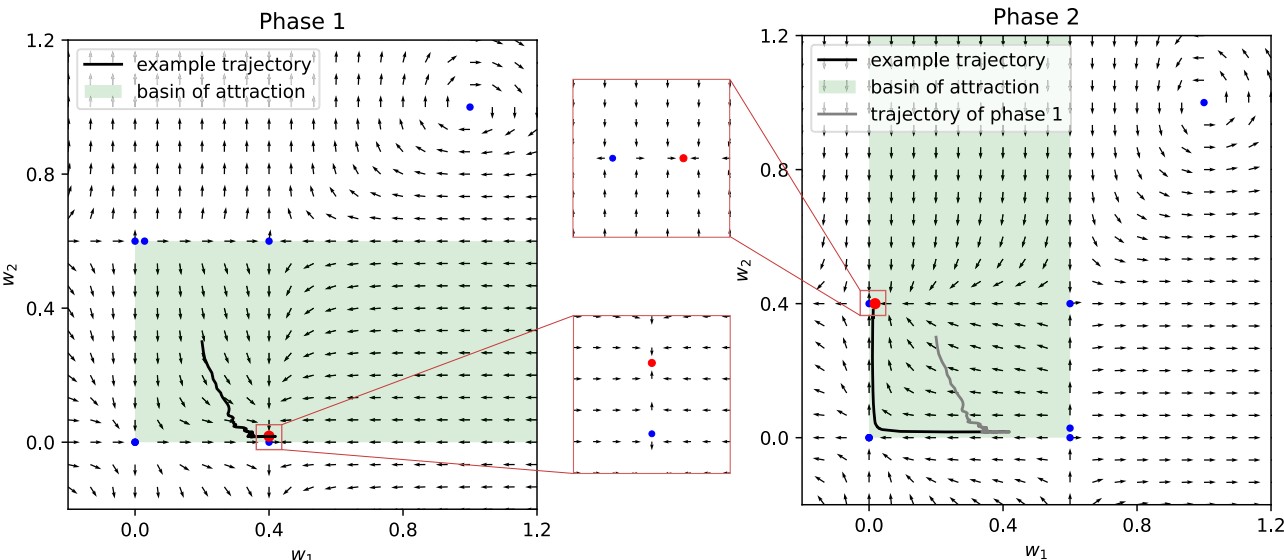

**Fig. 6 | Stability of the R-BCM system.** The stationary points (Eq.(15)) and the average vector field (Eq.(11)) are plotted in the weight space, during the first phase of learning (left, $R_1 = -1$ and $R_2 = 1.5$) and during the second phase (right, the values of $R_i$ are switched). The blue dots denote unstable stationary points, the red dots denote stable stationary points. The average vector field $E[\Delta\mathbf{w}]$ reveals that there is only one stable point (insets). A saddle point is present close to the stable node, delimiting its basin of attraction to non-negative values. The green area denotes the basin of attraction of the stable point. The stability properties are checked also through simulations with

multiple initial conditions. An example trajectory is plotted, showing that the weights tend towards the stable point during phase 1, and then correctly tend towards the new stable point after the switching. We note that the stationary points and the average vector field of phase 1 (before switching) and phase 2 (after switching) are related by a reflection across the identity line. It is important that the initial stable point falls inside the basin of attraction of the new stable point after the reflection. This is ensured by the fact that the positive $R_i$ is greater in magnitude than the negative one, as can be seen by substituting the values in Eq. (15).

transition matrix reads:

$$\Pi = \begin{pmatrix} p_{11} & p_{12} \\ p_{21} & p_{22} \end{pmatrix} = \begin{pmatrix} 1 - \max(0.05, y_1) & \max(0.05, y_2) \\ \max(0.05, y_1) & 1 - \max(0.05, y_2) \end{pmatrix} \quad (12)$$

The transition matrix is thus a function of the weight vector $\mathbf{w}$ (through $y_i = \mathbf{w} \cdot \mathbf{x}_i$). However, since we are looking for the stationary weight vectors, we assume the transition matrix to be constant. In this case, the probability vector $\mathbf{p}$ will converge to the eigenvector of $\Pi$ with eigenvalue 1, which is:

$$(p_1, p_2) = \left( \frac{p_{12}}{p_{12} + p_{21}}, \frac{p_{21}}{p_{12} + p_{21}} \right) \quad (13)$$

The stationary points are computed by solving the equation $E[\Delta\mathbf{w}] = \mathbf{0}$. Assuming that the input vectors $\mathbf{x}_1$ and $\mathbf{x}_2$ are linearly independent (as it was in our simulations), this leads back to solving two equations independently:

$$\begin{aligned} (p_{11}R_1 + p_{21}R_2)y_1(y_1 - \theta) &= 0 \\ (p_{22}R_2 + p_{12}R_1)y_2(y_2 - \theta) &= 0 \end{aligned} \quad (14)$$

We obtain 7 distinct stationary points:

$$\left\{ (0,0), (1,1), \left( \frac{R_1}{R_1 - R_2}, \frac{R_2}{R_2 - R_1} \right) \left( 0, \frac{R_2}{R_2 - R_1} \right), \left( \frac{R_1}{R_1 - R_2}, 0 \right), \right.$$
$$\left. \left( \theta, \frac{R_2}{R_2 - R_1} \right), \left( \frac{R_1}{R_1 - R_2}, \theta \right) \right\} \quad (15)$$

where the value of $\theta$ is the determined by the equation (valid in the "slow-learning" approximation[53], i.e. $\tau_w \gg \tau_\theta$):

$$\theta = E[y^2] = p_1 y_1^2 + p_2 y_2^2 \quad (16)$$

where $p_1$ and $p_2$ are given by Eq. (13). We note that the coincidence of the stationary points of the deterministic equation (11) with the stationary points of the stochastic equation (6a) has to be proven, by showing that the solutions converge. This was done analytically for the BCM equation[54], but doing the same for the R-BCM is non-trivial. Instead, we check the coincidence of the stationary points through numerical simulations. We use numerical simulations also to check the stability properties of the points and find that only one is stable, according to what we expected by looking at the average vector field (Fig. 6). The stable point is:

$$(w_1, w_2) = \begin{cases} \left( \frac{R_1}{R_1 - R_2}, \theta \right), & \text{if } R_1 = -1, R_2 = 1.5 \\ \left( \theta, \frac{R_2}{R_2 - R_1} \right), & \text{if } R_1 = 1.5, R_2 = -1 \end{cases} \quad (17)$$

Importantly, as in the original BCM, this is a *selective* point, meaning that the activity $y_i = \mathbf{w} \cdot \mathbf{x}_i$ is high for one input and low for the other.

**Parameters**

For the time constants of the system (10), we set $\tau_w = 100$ and $\tau_\theta = 50$, with intrinsic time units. Numerical simulations show that it must be $\tau_w \gg 1$ in order for the stationary points of Eq. (17) to be stable. This is due to the fact that the value of $R$ in Eq. (10) changes at every time step $\Delta t = 1$, and the stability of the system relies on an averaging operation over the $R$ (see Eq. (14)). Concerning $\tau_\theta$, the derivation of the stationary points in the previous section is valid under the assumption $\tau_w \gg \tau_\theta$ so that Eq. (16) holds. However numerical simulations show that a stable point exists also for greater values of $\tau_\theta$. Indeed, a stable point exists also when $\tau_\theta \to +\infty$, which corresponds to the case of the mutant mice. In this case, however, the stable point is not given anymore by Eq. (16). Numerical simulations show that the point is very close to the original one.

The function $R$ can assume only two values, $R_1 = R(S^1)$ and $R_2 = R(S^2)$. During the first training phase, we set $R_1 = 1.5$ and $R_2 = -1$; the values are

switched during the second phase. With these values, the stable steady state is such that the agent choices tend to avoid the state $S_i$ with $R_i = -1$. Let us mention that the need for having a positive signal is due to the form of the R-BCM equation (7): if the value of $R$ for the unpunished state is set to zero, the corresponding weight will not be able to increase after the switch from phase one to phase two. One might thus be tempted to interpret $-1$ as a negative reinforcement and 1.5 as a positive or neutral reinforcement. However, it is important to be aware that such interpretation is arbitrary: the avoidant behavior is determined by the *combination* of the two values. The fact that the values are opposite in sign ensures that the stationary points in (15) lay in the positive quadrant. The fact that the positive value is greater in magnitude than the negative value, ensures that, immediately after the switching, the system is in the basin of attraction of the new stable point (see Fig. 6).

When simulating the disruption of the D-serine feedback loop in mutant mice, the LTP threshold is fixed at $\theta_0 = 0.02$. This value corresponds to the average stationary value of the threshold obtained with normal mice, far away from the switching time.

### Reporting summary

Further information on research design is available in the Nature Portfolio Reporting Summary linked to this article.

### Data availability

No new datasets were generated. Any experimental data shown was replotted by the authors of this article using visual information contained in figures from ref. 32 and ref. 31. The corresponding citations of data sources are included as "Figures adapted from" in this article.

### Code availability

All the code used to simulate the experiments is publicly available(at githubhttps://github.com/CompNeuroTchuGroup/TheDserineHypothesis) with https://doi.org/10.5281/zenodo.115658701. An implementation of the model and visualization of the experimental outcomes in `Netlogo`[55] can be found at: https://verzep.github.io/R-BCM.html.

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

## Acknowledgements
This research was supported by the University of Bonn Medical Center, the Deutsche Forschungsgemeinschaft (DFG, German Research Foundation) - Project-ID 227953431 - SFB 1089 (P.V., C.W.C., D.M.K., T.T., C.H.), Joachim Herz Foundation (CWC), and the iBehave Network, which is funded by the Ministry of Culture and Science of the State of North Rhine-Westphalia (L.S., C.H., T.T.).

## Author contributions
All the authors (L.S., C.W., D.M., K.B., C.H., P.V., T.T.) conceived the study and initiated the work. L.S., P.V., T.T. formalized the mathematics of the model and designed the task. L.S. simulated the model and analyzed the data. All authors (L.S., C.W., D.M., K.B., C.H., P.V., T.T.) contributed to the manuscript and participated in the scientific discussion of the results.

## Funding

## Competing interests
The authors declare no competing interests.
