## [Peer review file · Communications Biology]

Reviewers' comments:

Reviewer #1 (Remarks to the Author):

The paper presents a biophysical mechanism for the classical BCM learning rule. Based on experimental data, the authors propose that an astrocytic D-serine acts to modulate the dynamically changing threshold for potentiation/ depotentiation and develop a functional models of how D-serine levels also depend on the postsynaptic activity. The authors choose to shape of the D-serine dependence on the postsynaptic activity to match the experimentally observed u-shape. The authors then show how this biophysically-based models, augmented with a reward-gating can account for learning and transient impairment of a place avoidance task in wild-type and genetically modified mice. Consistent with the experimental data, an agent based on the proposed rule learns to decrease probability of being in the punished state. The agent also learns to change the punished state location. In an agent where the D-serine levels are no longer dependent on the postsynaptic activity learning of the new location to avoid is delayed. This is also observed in mice.

The paper is clearly written and the model presented is elegant in its clarity. I think it provides clear predictions for experiments that would be valuable to understand how interactions between neurons and glia modulate and control plasticity and learning. There are several aspects of the paper that should be improved in order for it to be fit for publication.

I think the main issue is the lay out and the flow of the paper. At the moment, the material is presented in a rather disjoint manner: in the first part a local synaptic rule is laid out based on detailed data about the influence of D-serine, an NMDA co-agonist, on the amplitude of dendritic spikes. Then in the second part a behavioral task is modelled. This requires not only a large conceptual leap from single dendrite/single neuron to behavior of an animal. In order to model the task an augmented, reward-dependent learning rule is proposed. This new rule is central to the paper, but it is not really clearly discussed until the methods section. I believe it would be much better to move up the derivation/reasoning of the reward-dependent rule, including equations and implementation choices, to the main body of the paper. In the first part of the paper the authors did a very nice job laying out how they put together the BCM-like rule, I think they should do the same in the second part of the paper for the reward-modulated rule. I believe this would make the paper much clearer. Stability analysis, parameter selection, etc can be left in the methods.

I further have a number of questions about the models that should be addressed (note these are not in order of importance):

1. How are the parameters of the models picked? Seems like some of the functional forms are picked partially based on data and partially from modelling reasoning. But we are not really told

how the parameters are arrived at? Furthermore, there is only limited parametric exploration to explore the robustness of the model stability.

2. For the reward-dependent model, I am not sure it is best to refer to the choice unit as a “neuron”. Surely the behavior of the animal cannot be explained by plasticity of one neuron. Perhaps calling it a “choice unit”, to allow the possibility that it might represent a network or a population of neurons, might be better. Same goes for the phrasing related to “navigation” by the agent – strictly speaking there is no navigation of the environment in the model – the model just gives a probability of changing between two states.

3. Why is the non-punished state positively reinforced? What is the basis for that? Any biological/behavioral evidence for such? The authors mention that it is necessary for the model, but do not really explain why.

4. In the temporal profile of the behavior after the switch (Fig 4 top panel) there is a curious shoulder for the simulated behavior but not in the weights evolution? Where does it come from?

5. How are the amplitudes for the reward signals chosen? Do they all lead to same stability properties as shown in figure 6? We see that the learning impediment goes away with increased negative reward after switching and completely goes away as the negative reward for S1 after switch matches the positive reward for S1 that was there before the switch (let’s hope I got this right). Why is that? Can the authors comment on this? Does this imply that if one wanted to model the reversal-learning impediment independent of the punishment strength, this could be done by also increasing the positive reward for the non-punished state?

6. How are the weights and states initialized? Perhaps the authors stated it someplace in the paper, but I missed it?

7. Why the authors picked the functional form for the transition probability? My the max function and where did the value 0.05 come from?

8. I think the authors should give a more detailed derivation of the stability for the R-BCM model. The sketch they give is not sufficient without having to look up the referenced papers. I think this manuscript should be self-sufficient.

9. Looking at Figure 6, I see that the stable point is very close to an unstable point – I am trying to understand what that would mean? Does it mean that small perturbations in the weights would lead to an instability in the learning? How do these two points move in this phase space dependent on the model parameters? E.g. on the values of the rewards? By the way looks like the unstable point is a saddle – is that right?

10. The output of the model gives probability of switching. This makes a prediction that such activity should be observed in the hippocampus (or another structure that controls this behavior). Had such activity been observed? If not this might be another prediction of the model.

11. I think the authors should proofread the paper carefully– there are typos, especially in the figure legends.

Reviewer #2 (Remarks to the Author):

In this paper the authors study synaptic plasticity using a model that includes Serine release.

They map the resulting model to a reward modulated BCM model and are then able to explain experimental data.

In general I am a proponent of these combined modeling and data papers, as there is still a lot to discover about the precise details of synaptic plasticity rules and their biophysical implementation.

That said, I found the model definition rather confusing.

d is introduced as the serine concentration relative to a background level, but as can be inferred later ($d = -\theta$ and $D(y) = -y^2$), but never explicitly stated, d needs to be negative. So $[Ser]$ is high, unless y becomes active.

I suggest to include d_0 early on and state that $d < 0$. This would be further clarified by plotting θ and d time courses, for instance in figure 4.

Another issue that is ignored is that $[Ser] = d_0 - d$ should never be able to go below zero.

$D(y) = -y^2$ is used as a model for the production of serine, using Fig.1D as justification.

This is probably the weakest part of the model. The biggest effect (ie. highest $[ser]$ in the figure is at intermediate frequencies). So D should be something like $D = -(y - y_0)^2$, where y_0 is some 10Hz.

(I think it further requires that this requires the astrocytes being active, which was not clear to be the case in [28].)

I wonder why the authors are trying so hard to recover the BCM model; It might be much more interesting to model the data as is, e.g. $D(y) \propto -(y - y_0)^2$, and see what happens.

The rest of the paper looks reasonable, but without the obstacles cleared it did not make much sense to go into detail on the results of the model.

Reviewer #3 (Remarks to the Author):

The manuscript describes a possible biophysical mechanism for the Bienenstok-Cooper-Munro (BCM) synaptic plasticity rule, inspired by recently reported observations of a feedback loop between neural activity (in CA1 pyramidal cells) and astrocytic regulation that modulates the threshold and amplitude of dendritic spikes. The manuscript also proposes how the novel

biophysical interpretation of the BCM rule can be used to successfully model an otherwise enigmatic result from a behavioral learning experiment that blocked the action of astrocytes in the feedback loop.

The BCM synaptic plasticity rule is important in our understanding of the regulation of plasticity to achieve stability and specificity during neural learning. Consequently, the results, as claimed in the manuscript, would be an important contribution to the understanding of the mechanisms underlying the regulation of synaptic. Further, an ability to explain an unusual learning deficit would, based on the new interpretation of the BCM rule would provide an important link between plasticity and behaviour.

However, there are several issues with proposed interpretation of the BCM rule in terms of the neuro-astrocyte feedback that would need be resolved before the model is plausible (see below). Further, it was unclear to me how the model was able to successfully explain a key aspect of the learning deficit experiment.

Problems with the interpretation of the model.

1. The key variables in the model are the neural activity, y , and the extracellular D-serine concentration, d . The dynamic equation for the latter is $\dot{d} = -1/\tau (d - D(y))$. Here $D(y)$ is the equilibrium concentration of D-serine for a given level of neural activity. The authors argue that a good choice of this function $D(y) = -y^2$, based on recent experimental findings (Bohmbach et al (2022, ref [28])). However, in this case $\dot{d} = -1/\tau (d + y^2) < 0$. So the concentration of D-serine will always decrease, given that $d > 0$ (since it is a concentration). The authors mention in a figure caption that $d < 0$, and should be interpreted as the value of d relative to a maximum amount. However, it would be better to make this explicit in the mathematics. Also, the units make no sense: concentration is not measured in $(\text{Hz})^2$.

2. The choice for $D(y) = -y^2$ is based on the finding of Bohmbach et al (2022, ref [28]), particularly figure 4e that shows the amplitude of the slow component of the dendritic spike in CA1 pyramidal neurons following alveus stimulation at different frequencies. The amplitude is maximal at a stimulation frequency of 10 Hz, lower at 20 Hz, and very low at 4 and 40 Hz. The frequency of alveus stimulation is meant to be a proxy for neural activity, y , in these neurons. However, it is not clear why a function with a maximum at 10 Hz should be modelled with $-y^2$, which has a maximum at 0. One would expect a term like $-(y - y_0)^2$, where y_0 corresponds to 10 Hz. Furthermore, $D(y)$ is once again negative when it is an (equilibrium) concentration and so should be positive. Further, why is it reasonable to take the amplitude of the slow component of the dendritic spike as a proxy for equilibrium concentration of D-serine. The logic behind this should be made more explicit.

3. Later in the manuscript when the effect of astrocytic knock-out is being modelled the function $D(y)$ is changed to $D(y) = d_0$. I would think this should be the value obtained when there is no activity, so the original function $D(y=0) = d_0$. This implies $d_0 = 0 (=0^2)$.

4. In the BCM rule, the threshold, θ , should be greater than zero, but here $\theta = -d < 0$ for a concentration $d > 0$.

Mostly likely these issues can be resolved by adjusting the functions appropriately, but should be made explicit.

Problems with the simulation of the learning deficit in mutant mice:

In the simulation of behavior in Figure 4, it is not clear why the mutant and control mice should show exactly the same evolution for the synaptic weight and rate of visits to site 1 prior to switching the adverse stimuli to site 2. Surely in the mutant case, if $D(y) = d_0$ instead of $D(y) = -y^2$, there must be an effect on the weight evolution according to equation (1). In the manuscript this is simply glossed over stating “The same behavioral pattern is observed when imposing condition (4), simulating the disruption of the D-serine feedback loop in mutant mice.

Finally, in Table 1 is stated that $\Delta w_k = 0$ if the other weight change is non-zero. It is not clear why this should be - it does not seem to be reflected in e.g. Eq. 7a.

Dear Dr. Palminteri, dear reviewers

Thank you for the helpful comments and the opportunity to resubmit a revised version of our manuscript. Below we address all comments point-by-point. The referee comments are in **bold font** while our answers are in regular font. The manuscript parts that have been adapted in response to referee feedback are highlighted using **blue font** and we use the same convention here when quoting them.

Reviewer 1

The paper presents a biophysical mechanism for the classical BCM learning rule. Based on experimental data, the authors propose that an astrocytic D-serine acts to modulate the dynamically changing threshold for potentiation/depotentiation and develop a functional model of how D-serine levels also depend on the postsynaptic activity. The authors choose to shape of the D-serine dependence on the postsynaptic activity to match the experimentally observed u-shape. The authors then show how this biophysically-based models, augmented with a reward-gating can account for learning and transient impairment of a place avoidance task in wild-type and genetically modified mice. Consistent with the experimental data, an agent based on the proposed rule learns to decrease probability of being in the punished state. The agent also learns to change the punished state location. In an agent where the D-serine levels are no longer dependent on the postsynaptic activity learning of the new location to avoid is delayed. This is also observed in mice.

The paper is clearly written and the model presented is elegant in its clarity. I think it provides clear predictions for experiments that would be valuable to understand how interactions between neurons and glia modulate and control plasticity and learning. There are several aspects of the paper that should be improved in order for it to be fit for publication.

Main issue

I think the main issue is the lay out and the flow of the paper. At the moment, the material is presented in a rather disjoint manner: in the first part a local synaptic rule is laid out based on detailed data about the influence of D-serine, an NMDA co-agonist, on the amplitude of dendritic spikes. Then in the second part a behavioral task is modelled. This requires not only a large conceptual leap from single dendrite/single neuron to behavior of an animal. In order to model the task an augmented, reward-dependent learning rule is proposed. This new rule is central to the paper, but it is not really clearly discussed until the methods section. I believe it would be much better to move up the derivation/reasoning of the reward-dependent rule, including equations and implementation choices, to the main body of the paper. In the first part of the paper the authors did a very nice job laying out how the put together the BCM-like rule, I think they should do the same in the second part of the paper for the reward-modulated rule. I believe this would make the paper much clearer. Stability analysis, parameter selection, etc can be left in the methods.

Thank you for the suggestion to reorder the methods and results within our manuscript. We have followed this suggestion also for the second part of the manuscript. We agree this new layout is indeed now much more accessible for readers. We thank the reviewer for the useful suggestion which we have implemented in the attached, revised version of the manuscript.

Specific questions

I further have a number of questions about the models that should be addressed (note these are not in order of importance):

1. **How are the parameters of the models picked?** Seems like some of the functional forms are picked partially based on data and partially from modelling reasoning. But we are not really told how the parameters are arrived at? Furthermore, there is only limited parametric exploration to explore the robustness of the model stability.

Thank you for this helpful comment. We have restructured the Methods section and extended it incorporating these suggestions of the referee. We added a section named 'Parameters', where we introduce and define all parameter values and comment on them. We also comment on how the parameters' choice affects model stability. Here is the new text:

For the time constants of the system (Eq. 10), we set $\tau_w = 100$ and $\tau_\theta = 50$, with intrinsic time units. Our numerical simulations show that $\tau_w \gg 1$ is needed in order for the stationary points of Eq. 17 to be stable. This is due to the fact that the value of R in Eq. 10 changes at every time step $\Delta t = 1$, and the stability of the system relies on an averaging operation over the R (see Eq. 14). Concerning τ_θ , the derivation of the stationary points in the previous section is valid under the assumption $\tau_w \gg \tau_\theta$, so that Eq. 16 holds. However numerical simulations show that a stable point exists also for greater values of τ_θ . Indeed, a stable point exists also when $\tau_\theta \rightarrow +\infty$, which corresponding to the case of the mutant mice. In this case, however, the stable point is not given anymore by Eq.16. However, our numerical simulations show that the point is very close to the original one.

The function R can assume only two values, $R_1 = R(S^1)$ and $R_2 = R(S^2)$. During the first phase of training, we set $R_1 = 1.5$ and $R_2 = -1$; the values are switched during the second phase. With this values, the stable steady state is such that the agent choices tend to avoid the state S_i with $R_i = -1$. One might thus be tempted to interpret -1 as a negative reinforcement and 1.5 as a positive or neutral reinforcement. However, it is important to be aware that such interpretation is arbitrary: the avoidant behavior is determined by the *combination* of the two values. The fact that the values are opposite in sign ensures that the stationary points in (15) lay in the positive quadrant. The fact that the positive value is greater in magnitude than the negative value ensures that, immediately after the switching, the system is in the basin of attraction of the new stable point (see Fig. 6).

When simulating the disruption of the D-serine feedback loop in mutant mice, the LTP threshold is fixed at $\theta_0 = 0.02$. This value corresponds to the average stationary value of the threshold obtained by simulating the normal mice, far away from the switching time.

2. **For the reward-dependent model, I am not sure it is best to refer to the choice unit as a “neuron”. Surely the behavior of the animal cannot be explained by plasticity of one neuron. Perhaps calling it a “choice unit”, to allow the possibility that it might represent a network or a population of neurons, might be better. Same goes for the phrasing related to “navigation” by the agent – strictly speaking there is no navigation of the environment in the model – the model just gives a probability of changing between two states.**

We rephrased the related parts. We substituted the term "neuron" with "action unit" and the term "navigation" with "movement" or "change of state".

3. **Why is the non-punished state positively reinforced? What is the basis for that? Any biological/behavioral evidence for such? The authors mention that it is necessary for the model, but do not really explain why.**

While the presence of positive and negative values naturally leads to interpreting them as reward and punishment. Yet such intuitive interpretation has also an arbitrary component, since it is the combination or the difference between the two values which ultimately determines the behavior not the individual values themselves. We commented on this interpretation issue in the Parameters section of Methods

The function R can assume only two values, $R_1 = R(S^1)$ and $R_2 = R(S^2)$. During the first phase of training, we set $R_1 = 1.5$ and $R_2 = -1$; the values are switched during the second phase. With this values, the stable steady state is such that the agent choices tend to avoid the state S_i with $R_i = -1$. One might thus be tempted to interpret $R = -1$ as a negative reinforcement and $R = 1.5$ as positive or neutral reinforcement. However, it is important to be aware that such interpretation is arbitrary: the avoidant behavior is determined by the *combination* of the two values. The fact that the values are opposite in sign ensures that the stationary points in (15) lay in the positive quadrant. The fact that the positive value is greater in magnitude than the negative value, ensures that, immediately after the switching, the system is in the basin of attraction of the new stable point (see Fig. 6).

4. **In the temporal profile of the behavior after the switch (Fig 4 top panel) there is a curious shoulder for the simulated behavior but not in the weights evolution? Where does it come from?**

The shoulder profile is due to the fact that the probability of being in State 1 is the sum of two terms: the probability of changing when the agent is in State 2 and the probability of staying when the agent is in State 1.

The first probability coincides with the value of w_2 , which rapidly falls down after switching, producing the first drop in the overall probability. The second probability is given by $1 - w_1$, and the rising of w_1 is consistently delayed in the mutant mice, generating a plateau before the final decrease.

5. **How are the amplitudes for the reward signals chosen? Do they all lead to same stability properties as shown in figure 6? We see that the learning impediment goes away with an increased negative reward after switching and completely goes away as the negative reward for S1 after switch matches the positive reward for S1 that was there before the switch (let’s hope I got this right). Why is that? Can the authors comment on this? Does this imply that if one wanted to model the reversal-learning impediment independent of the punishment strength, this could be done by also increasing the positive reward for the non-punished state?**

We thank again the reviewer for this insightful comment. We have addressed the first two questions (choice of the reward signals and their relation to stability) addressing point 1 of the review above. To answer the second question, we plot below how the learning curves for the mutant mice change as we systematically change the values of R_1 (negative) and R_2 (positive). The first figure is the same plot as Fig.5 (right), where the magnitude of R_1 is progressively increased, and the mutant mice learn progressively faster. The second figure shows the opposite behavior as we progressively decrease the magnitude of R_1 . The third and fourth figures show that varying the magnitude of R_2 , the following behavior occurs: increasing (decreasing) the magnitude of R_2 , the mutant mice learn slower (faster). Thus, the answer to the question is that indeed it is possible to modulate the learning impediment independent of the negative value of R but change the positive value. This is not surprising since it is the combination of the two values that determines the avoidance behavior, as commented in the answer to point 3) of the review.

6. **How are the weights and states initialized? Perhaps the authors stated it someplace in the paper, but I missed it?**

Thank you for this comment, a clear statement was indeed missing, and we added it in the Methods section:

The initial values of the weights are $(w_1, w_2) = (0.2, 0.2)$, and the agent starts from a random state.

7. **Why the authors picked the functional form for the transition probability? My the max function and where did the value 0.05 come from?**

We set the transition probability to be linearly proportional to the unit activity because it was the simplest choice. We set 0.05 as the minimum value in order to maintain a minimum level of exploration. We added this motivation in the Methods section when introducing the equation:

The mouse switches state at the next time step with a probability p_{change} directly proportional to the firing rate of the action unit, with a minimum value of 0.05:

$$p_{\text{change}} = \max(0.05, y_t). \quad (9)$$

The lower bound of p_{change} was introduced to maintain a minimum level of exploration.

8. **I think the authors should give a more detailed derivation of the stability for the R-BCM model. The sketch they give is not sufficient without having to look up the referenced papers. I think this manuscript should be self-sufficient**

We extended our stability analysis in the Methods section. We also revised the explanatory figure, which now highlights the basin of attraction of the stable point, we compare how it changes from the first to the second phase of learning, and we show an example trajectory. Below, we report the new text.

We compute the stationary points of R-BCM with the same procedure used for obtaining the stationary points of the original BCM [8, 50], i.e. replacing the stochastic differential equation (7) with the same equation averaged over the input environment:

$$E[\Delta \vec{w}] = -p_1 \vec{x}_1 (p_{11} R_1 + p_{21} R_2) y_1 (y_1 - \theta) - p_2 \vec{x}_2 (p_{11} R_1 + p_{21} R_2) y_1 (y_1 - \theta) \quad (11)$$

where $y_1 = \vec{w} \cdot \vec{x}_1$, $y_2 = \vec{w} \cdot \vec{x}_2$, p_i is the probability of being in state i , and p_{ij} is the probability of the transition from state j to state i . The dynamics of the agent can be described as a Markov chain over the two-state space, so that the probabilities $\vec{p} = (p_1, p_2)$ obey $\vec{p}(t+1) = \Pi \vec{p}(t)$, where Π is the transition matrix. We recall that the probability of switching state is related to the firing rate of the action unit through Eq. 9. Thus, for a fixed weight vector, the transition matrix reads:

$$\Pi = \begin{pmatrix} p_{11} & p_{12} \\ p_{21} & p_{22} \end{pmatrix} = \begin{pmatrix} 1 - \max(0.05, y_1) & \max(0.05, y_2) \\ \max(0.05, y_1) & 1 - \max(0.05, y_2) \end{pmatrix} \quad (12)$$

The transition matrix is thus a function of the weight vector \vec{w} (through $y_i = \vec{w} \cdot \vec{x}_i$). However, since we are looking for the stationary weight vectors, we assume the transition matrix to be constant. In this case, the probability vector \vec{p} will converge to the eigenvector of Π with eigenvalue 1, which is:

$$(p_1, p_2) = \left(\frac{p_{12}}{p_{12} + p_{21}}, \frac{p_{21}}{p_{12} + p_{21}} \right) \quad (13)$$

The stationary points are computed by solving the equation $E[\Delta \vec{w}] = \vec{0}$. Assuming that the input vectors \vec{x}_1 and \vec{x}_2 are linearly independent (as it was in our simulations), this leads back to solving two equations independently:

$$\begin{aligned} (p_{11} R_1 + p_{21} R_2) y_1 (y_1 - \theta) &= 0 \\ (p_{22} R_2 + p_{12} R_1) y_2 (y_2 - \theta) &= 0 \end{aligned} \quad (14)$$

We obtain 7 distinct stationary points:

$$\left\{ (0, 0), (1, 1), \left(\frac{R_1}{R_1 - R_2}, \frac{R_2}{R_2 - R_1} \right), \left(0, \frac{R_2}{R_2 - R_1} \right), \left(\frac{R_1}{R_1 - R_2}, 0 \right), \left(\theta, \frac{R_2}{R_2 - R_1} \right), \left(\frac{R_1}{R_1 - R_2}, \theta \right) \right\} \quad (15)$$

where the value of θ is the determined by the equation (valid in the ‘‘slow-learning’’ approximation [50], i.e. $\tau_w > \tau_\theta$):

$$\theta = E[y^2] = p_1 y_1^2 + p_2 y_2^2 \quad (16)$$

where p_1 and p_2 are given by Eq.13. We note that the coincidence of the stationary points of the deterministic equation 11 with the stationary points of the stochastic equation 6a has to be proven, by showing that the solutions converge. This was done analytically for the BCM equation [51], but doing the same for the R-BCM

Figure 6: **Stability of the R-BCM system.** The stationary points (Eq. 15) and the average vector field (Eq.11) are plotted in the weight space, during the first phase of learning (left, $R_1 = -1$ and $R_2 = 1.5$) and during the second phase (right, the values of R_i are switched). The blue dots denote unstable stationary points, the red dots denote stable stationary points. The average vector field $E[\Delta \vec{w}]$ reveals that there is only one stable point (insets). The green area denotes the basin of attraction of the stable point. The stability properties are checked also through simulations with multiple initial conditions. An example trajectory is plotted, showing that the weights tend towards the stable point during phase 1, and then correctly tend towards the new stable point after the switching. We note that the stationary points and the average vector field of phase 1 (before switching) and phase 2 (after switching) are related by a reflection across the identity line. It is important that the initial stable point falls inside the basin of attraction of the new stable point after the reflection. This is ensured by the fact that the positive R_i is greater in magnitude than the negative one, as can be seen by substituting the values in Eq. 15.

is non-trivial. Instead, we check the coincidence of the stationary points through numerical simulations. We use numerical simulations also to check the stability properties of the points, and find that only one is stable, according to what we expected by looking at the average vector field (Fig. 6). We use numerical simulations also to check the stability properties of the points, and find that only one is stable. Importantly, as in the original BCM, this is a *selective* point, meaning that the activity $y_i = \vec{w} \cdot \vec{x}_i$ is high for one input and low for the other.

9. **Looking at Figure 6, I see that the stable point is very close to an unstable point – I am trying to understand what that would mean? Does it mean that small perturbations in the weights would lead to an instability in the learning? How do these two points move in this phase space dependent on the model parameters? E.g. on the values of the rewards? By the way looks like the unstable point is a saddle – is that right?**

We highlighted the basin of attraction of the stable point in the new figure (Fig. 6). Yes, the stable point is very close to a saddle point, which restrict the basin of attraction on one side. The relation between the location of the stable point and the saddle point can be deduced by the stability analysis we presented. Consider the Phase 1, where the stable point is $\vec{w} = (R_1/(R_1 - R_2), \theta)$ and the saddle point is $\vec{w} = (R_1/(R_1 - R_2), 0)$. The points have the same x-coordinate regardless of the choice of R_1, R_2 . The y-coordinate of the saddle point is always 0, while the y-coordinate of the stable point is given by the solution of Eq. 16 (reported in the answer the previous point). For the $R_1 = -1, R_2 = 1.5$ it is $\theta \approx 0.02$. Perturbations of such magnitude would make the system unstable. We recognize that this represent a weakness of the model. However, due to the fact that the y-coordinate of the saddle point is 0, we could solve the instability by constraining the weight to be non-negative (in analogy with an excitatory synapse which cannot become inhibitory).

10. **The output of the model gives probability of switching. This makes a prediction that such activity should be observed in the hippocampus (or another structure that controls this behavior). Had such activity been observed? If not this might be another prediction of the model.**

This is an interesting point. It is true that the model predicts that astrocytic D-serine signaling plus a reward signal at hippocampal CA3-CA1 synapses would create CA1 cells whose activity could predict the switching probability. However, we think that the model is too simplified with respect to the brain structure to sustain such a prediction.

11. **I think the authors should proofread the paper carefully– there are typos, especially in the figure legends.**

Typos and missing references were corrected.

Reviewer 2

In this paper the authors study synaptic plasticity using a model that includes Serine release. They map the resulting model to a reward modulated BCM model and are then able to explain experimental data. In general I am a proponent of these combined modeling and data papers, as there is still a lot to discover about the precise details of synaptic plasticity rules and their biophysical implementation.

That said, I found the model definition rather confusing. d is introduced as the serine concentration relative to a background level, but as can be inferred later ($d = -\theta$ and $D(y) = -y^2$), but never explicitly stated, d needs to be negative. So $[Ser]$ is high, unless y becomes active. I suggest to include d_0 early on and state that $d < 0$. This would be further clarified by plotting θ and d time courses, for instance in figure 4. Another issue that is ignored is that $[Ser] = d_0 - d$ should never be able to go below zero. $D(y) = -y^2$ is used as a model for the production of serine, using Fig.1D as justification. This is probably the weakest part of the model. The biggest effect (ie. highest $[ser]$ in the figure is at intermediate frequencies). So D should be something like $D = -(y - y_0)^2$, where y_0 is some $10Hz$.

We would like to thank the reviewer for the useful comments. We addressed the reported issues with a substantial reformulation of the D-serine dynamics. The variable d now represents the absolute level of D-serine, and its stationary value for a constant activity is $D(y) = D_0 - (y - y_0)^2$, with a maximum of $D_0 > 0$ for $y = y_0$. We note that $D(y)$ can still become negative for values of y outside a certain interval I . To remove this effect, we could have introduced

an asymptotic behavior towards zero, but we preferred to keep the mathematical form simpler. It is always possible to choose D_0 and a so that the interval I covers all the range of physiological values for y . The new formulation is presented in the first subsection of the Results section, and we report it here:

The D-serine hypothesis: astrocytes orchestrate BCM plasticity

Recent experiments on mice CA1 hippocampal neurons have shed light on a novel molecular mechanism, forming a feedback loop through which the activity of pyramidal neurons influences their future dynamics and plasticity [28]. The feedback loop is mediated by astrocytes and is enacted by two main molecules: endocannabinoids released by neurons upon activation, which bind to the cannabinoid receptors (CBRs) of astrocytes [29, 30], and D-serine, released into the extracellular environment in response to astrocytic calcium signaling triggered by CBRs activation [14, 24, 31, 32] (Fig. 1a).

Let y represent the average firing rate of a neuron population, and d denote the concentration of D-serine in the extracellular environment. Based on previous observations, we describe the dynamics of d so that it follows a given function of the post-synaptic activity, denoted as $D(y)$, with a time constant τ_d :

$$\dot{d} = -\frac{1}{\tau_d}(d - D(y)) \quad (1)$$

The right-hand side of Eq. (1) can be interpreted as the sum of a D-serine degradation/uptake term and a term that describes activity-dependent D-serine supply. The dependence of the D-serine release on the post-synaptic activity has been studied by monitoring D-serine-dependent dendritic integration during the axonal stimulation of CA1 pyramidal neurons at different frequencies [28]. Notably, an unexpected bell-shaped dependence has been observed, with a peak at 10Hz (Fig. 1, d). Based on this, we choose the simple mathematical form:

$$D(y) = D_0 - a(y - y_0)^2 \quad (2)$$

where D_0 is the maximum level of D-serine, occurring when $y = y_0$, and a is a constant with the appropriate dimensions.

The importance of D-serine stems from its interaction with synaptic NMDAR, which plays a fundamental role in the induction of synaptic plasticity [33]. D-serine has been identified as an essential endogenous ligand for the glycine site of synaptic NMDARs in the brain [20, 21, 27]. Consequently, its presence is required for the activation of N-methyl-D-aspartate receptors (NMDARs). Consistent with these findings, alterations in D-serine levels have been unequivocally associated with changes in the activity-dependent nature of long-term synaptic modifications [27]. In particular, the same stimulation protocol induced LTP or long-term depression (LTD) depending on the level of D-serine. In the presence of a high (low) level of D-serine, LTP (LTD) is facilitated, and the same effect is equally seen with three different stimulation protocols. Such observation suggests that the effect of D-serine alterations can be interpreted as a shifting of a threshold for LTP induction (Fig. 1c). We denote with \vec{w} a vector of synaptic strengths and with \vec{x} a vector of pre-synaptic activities. We define the integration of the input vector with the synaptic strengths to track the neural activity with respect to the reference value y_0 , thus $z = y - y_0 = \vec{w} \cdot \vec{x}$. Finally, we define an update rule for the synaptic strengths in such a way that the direction of plasticity depends on the level of D-serine through a function $\theta(d)$:

$$\tau_w \dot{\vec{w}} = \vec{x} z (z - \theta(d)) \quad (3)$$

where τ_w is a time constant. The proportionality of the update to the pre- and post-synaptic firing realizes the Hebbian principle, while the quadratic dependence on the post-synaptic rate has been suggested by experimental measurements ([5, 27]) and has been derived from spikes-based learning rules (STDP [34, 35], calcium-based plasticity [36]). The equation is formally equivalent to the BCM update rule [1], with the important difference that here the threshold is no more a postulated dynamical variable but a function of the D-serine concentration. We show now that, choosing the function $\theta(d)$ according to experimental observations and considering the dynamics of D-serine described by Eqs. 1 and 2, we obtain a dynamical system equivalent to BCM.

According to the fact that a higher level of D-serine facilitates long-term potentiation (LTP), we choose the threshold function to be inversely proportional to the D-serine concentration:

$$\theta(d) = b(D_0 - d) \quad (4)$$

where b is a constant with the appropriate dimensions. The threshold is 0 when d is at its maximum value D_0 , and it increases as d decreases. We define the new variable $\theta = b(D_0 - d)$ ($\dot{\theta} = -bd$), and substitute it in the Eq.s 1 and 3, obtaining:

$$\begin{aligned} \tau_w \dot{\vec{w}} &= \vec{x}z(z - \theta) \\ \tau_d \dot{\theta} &= -\theta + ab \cdot z^2 \\ z &= \vec{w} \cdot \vec{x} \end{aligned} \quad (5)$$

which is equivalent to the BCM plasticity rule [1], and coincides exactly by setting $ab = 1$, as we do for the rest of this work.

(I think it further requires that this requires the astrocytes being active, which was not clear to be the case in [28].)

Astrocytic Ca^{2+} signaling, triggered by the activation of cannabinoid receptors, was measured and reported (see Fig.3b, Fig.4c in [28]).

I wonder why the authors are trying so hard to recover the BCM model; It might be much more interesting to model the data as is, e.g. $D(y) \propto -(y - y_0)^2$., and see what happens.

We certainly appreciate the suggestion. However, we chose to recover the BCM model for the following reasons. First, to deliver a simple and strong message. The similarities between the experimentally observed mechanism and the BCM postulate are striking, and the derivation of the very BCM from the description of such mechanism is the more direct way to show it. Second, to constrain the modeling of plasticity. Experimental data on the effect of D-serine on synaptic plasticity and on the regulation of D-serine are not very detailed yet; the "objective" of recovering the BCM equations added the constraints to guide the modeling choices. Nevertheless, more precise modeling is desirable in the future. Such modeling would require going deeper into the intricate biology of synaptic plasticity, starting from a calcium-based model of NMDAR-dependent plasticity rather than a rate-based model.

The rest of the paper looks reasonable, but without the obstacles cleared it did not make much sense to go into detail on the results of the model.

We thank the reviewer for the useful comments, and we welcome further feedback.

Reviewer 3

The manuscript describes a possible biophysical mechanism for the Bienenstock-Cooper-Munro (BCM) synaptic plasticity rule, inspired by recently reported observations of a feedback loop between neural activity (in CA1 pyramidal cells) and astrocytic regulation that modulates the threshold and amplitude of dendritic spikes. The manuscript also proposes how the novel biophysical interpretation of the BCM rule can be used to successfully model an otherwise enigmatic result from a behavioral learning experiment that blocked the action of astrocytes in the feedback loop. The BCM synaptic plasticity rule is important in our understanding of the regulation of plasticity to achieve stability and specificity during neural learning. Consequently, the results, as claimed in the manuscript, would be an important contribution to the understanding of the mechanisms underlying the regulation of synaptic. Further, an ability to explain an unusual learning deficit would, based on the new interpretation of the BCM rule would provide an important link between plasticity and behaviour.

However, there are several issues with proposed interpretation of the BCM rule in terms of the neuro-astrocyte feedback that would need be resolved before the model is plausible (see below). Further, it was unclear to me how the model was able to successfully explain a key aspect of the learning deficit experiment.

We thank the reviewer for the helpful comments. The problems raised with the interpretation of the model were all linked to the modeling of D-serine dynamics. We reformulated the modeling section with significant changes to incorporate the suggestions received. We think that the modeling of D-serine dynamics is now much clearer, and we would like to thank the reviewer for pointing at it. The reformulation concerns the first subsection of the Results section, which we report below. We then address the specific points raised by the reviewer one by one.

The D-serine hypothesis: astrocytes orchestrate BCM plasticity

Recent experiments on mice CA1 hippocampal neurons have shed light on a novel molecular mechanism, forming a feedback loop through which the activity of pyramidal neurons influences their future dynamics and plasticity [28]. The feedback loop is mediated by astrocytes and is enacted by two main molecules: endocannabinoids released by neurons upon activation, which bind to the cannabinoid receptors (CBRs) of astrocytes [29, 30], and gliotransmitter D-serine, released in the extracellular environment by astrocytes in response to the calcium signaling triggered by CBRs activation [14, 24, 31, 32] (Fig. 1a).

Let y represent the average firing rate of a neuron population, and d denote the concentration of D-serine in the extracellular environment. Based on previous observations, we describe the dynamics of d so that it follows a given function of the post-synaptic activity, denoted as $D(y)$, with a time constant τ_d :

$$\dot{d} = -\frac{1}{\tau_d} (d - D(y)) \quad (1)$$

The right-hand side of Eq. (1) can be interpreted as the sum of a D-serine degradation/uptake term and a term that describes astrocyte-dependent D-serine supply. The dependence of the D-serine release on the post-synaptic activity has been studied by monitoring D-serine-dependent dendritic integration during the axonal stimulation of CA1 pyramidal neurons at different frequencies [28]. Notably, an unexpected bell-shaped dependence has been observed, with a peak at 10Hz (Fig. 1, d). Based on this, we choose the simple mathematical form:

$$D(y) = D_0 - a(y - y_0)^2 \quad (2)$$

where D_0 is the maximum level of D-serine, occurring when $y = y_0$, and a is a constant with the appropriate dimensions.

The importance of D-serine stems from its interaction with synaptic NMDAR, which plays a fundamental role in the induction of synaptic plasticity [33]. D-serine has been identified as an essential endogenous ligand for the glycine site of synaptic NMDARs in the brain [20, 21, 27]. Consequently, its presence is required for the activation of NMDARs. Consistent with these findings, alterations in D-serine levels have been unequivocally associated with changes in the activity-dependent nature of long-term synaptic modifications [27]. In particular, the same stimulation protocol induced LTP or long-term depression (LTD) depending on the level of D-serine. In the presence of a high (low) level of D-serine, LTP (LTD) is facilitated, and the same effect is equally seen with three different stimulation protocols. Such observation suggests that the effect of D-serine alterations can be interpreted as a shifting of a threshold for LTP induction (Fig. 1c). We denote with \vec{w} a vector of synaptic strengths and with \vec{x} a vector of pre-synaptic activities. We define the integration of the input vector with the synaptic strengths to track the neural activity with respect to the reference value y_0 , thus $z = y - y_0 = \vec{w} \cdot \vec{x}$. Finally, we define an update rule for the synaptic strengths in such a way that the direction of plasticity depends on the level of D-serine through a function $\theta(d)$:

$$\tau_w \dot{\vec{w}} = \vec{x} z (z - \theta(d)) \quad (3)$$

where τ_w is a time constant. The proportionality of the update to the pre- and post-synaptic firing realizes the Hebbian principle, while the quadratic dependence on the post-synaptic rate has been suggested by experimental measurements

([5, 27]) and has been derived from spikes-based learning rules (STDP [34, 35], calcium-based plasticity [36]). The equation is formally equivalent to the BCM update rule [1], with the important difference that here the threshold is no more a postulated dynamical variable but a function of the D-serine concentration. We show now that, choosing the function $\theta(d)$ according to experimental observations and considering the dynamics of D-serine described by Eq.s 1 and 2, we obtain a dynamical system equivalent to BCM.

According to the fact that a higher level of D-serine facilitates LTP, we choose the threshold function to be inversely proportional to the D-serine concentration:

$$\theta(d) = b(D_0 - d) \quad (4)$$

where b is a constant with the appropriate dimensions. The threshold is 0 when d is at its maximum value D_0 , and it increases as d decreases. We define the new variable $\theta = b(D_0 - d)$ ($\dot{\theta} = -bd$), and substitute it in the Eq.s 1 and 3, obtaining:

$$\begin{aligned} \tau_w \dot{\vec{w}} &= \vec{x}z(z - \theta) \\ \tau_d \dot{\theta} &= -\theta + ab \cdot z^2 \\ z &= \vec{w} \cdot \vec{x} \end{aligned} \quad (5)$$

which is equivalent to the BCM plasticity rule [1], and coincides exactly by setting $ab = 1$, as we do for the rest of this work.

Problems with the interpretation of the model

1. **The key variables in the model are the neural activity, y , and the extracellular D-serine concentration, d . The dynamic equation for the latter is $\dot{d} = -1/\tau(d - D(y))$. Here $D(y)$ is the equilibrium concentration of D-serine for a given level of neural activity. The authors argue that a good choice of this function $D(y) = -y^2$, based on recent experimental findings (Bohmbach et al (2022, ref [28])). However, in this case $\dot{d} = -1/\tau(d + y^2) < 0$. So the concentration of D-serine will always decrease, given that $d > 0$ (since it is a concentration). The authors mention in a figure caption that $d < 0$, and should be interpreted as the value of d relative to a maximum amount. However, it would be better to make this explicit in the mathematics. Also, the units make no sense: concentration is not measured in $(Hz)^2$.**

In the new formulation, we start from a form of $D(y)$ more adherent to the biological interpretation, i.e., $D(y) = D_0 - a(y - y_0)^2$. Now $D(y)$ is positive within a certain interval I . It can still become negative for values of y outside that interval. However, it is always possible to choose D_0 and a so that the interval I covers all the range of physiological values for y . The conversion factor a resolves the units problem, converting the rate squared units to concentration units.

2. **The choice for $D(y) = -y^2$ is based on the finding of Bohmbach et al (2022, ref [28]), particularly figure 4e that shows the amplitude of the slow component of the dendritic spike in CA1 pyramidal neurons following alveus stimulation at different frequencies. The amplitude is maximal at a stimulation frequency of 10 Hz, lower at 20 Hz, and very low at 4 and 40 Hz. The frequency of alveus stimulation is meant to be a proxy for neural activity, y , in these neurons. However, it is not clear why a function with a maximum at 10 Hz should be modelled with $-y^2$, which has a maximum at 0. One would expect a term like $-(y - y_0)^2$, where y_0 corresponds to 10 Hz. Furthermore, $D(y)$ is once again negative when it is an (equilibrium) concentration and so should be positive. Further, why is it reasonable to take the amplitude of the slow component of the dendritic spike as a proxy for equilibrium concentration of D-serine. The logic behind this should be made more explicit.**

We would like to thank the reviewer for pointing to this. According to the comment, the new formulation starts from $D(y) = D_0 - a(y - y_0)^2$, so that the D-serine concentration has a maximum value D_0 for $y = y_0$, and it is no longer negative. As it regards the reason why the amplitude of the slow component of dendritic spike is a proxy for the D-serine concentration, this has been investigated in [28], where they measure such amplitude while manipulating the levels of D-serine.

3. **Later in the manuscript when the effect of astrocytic knock-out is being modelled the function $D(y)$ is changed to $D(y) = d_0$. I would think this should be the value obtained when there is no activity, so the original function $D(y = 0) = d_0$. This implies $d_0 = 0 (= 0^2)$.**

The fact that the value of D-serine in the absence of activity can be different from zero is explicit in the new formulation, where $D(y) = D_0 - a(y - y_0)^2$. The value obtained when there is no activity is now $D(0) = D_0 - ay_0^2$.

4. **In the BCM rule, the threshold, theta, should be greater than zero, but here $\theta = -d < 0$ for a concentration $d > 0$. Mostly likely these issues can be resolved by adjusting the functions appropriately, but should be made explicit.**

The issues have been resolved by defining $\theta(d) = b(D_0 - d)$, where b is a conversion factor and D_0 is the maximum value of D-serine. Thus, $\theta > 0$ for all possible d .

Problems with the simulation of the learning deficit in mutant mice

In the simulation of behavior in Figure 4, it is not clear why the mutant and control mice should show exactly the same evolution for the synaptic weight and rate of visits to site 1 prior to switching the adverse stimuli to site 2. Surely in the mutant case, if $D(y) = d_0$ instead of $D(y) = -y^2$, there must be an effect on the weight evolution according to equation (1). In the manuscript this is simply glossed over stating “The same behavioral pattern is observed when imposing condition (4), simulating the disruption of the D-serine feedback loop in mutant mice.

This is the aspect that makes the experimental observations so intriguing and hard to explain: why does the disruption of the D-serine feedback loop impair learning, but only in the second phase when the mouse has to “reverse/unlearn” a previously learned behavior? In our simulation, the key difference between the two phases of learning is the starting value of the weights: in the first phase, they start from an intermediate value, while in the second phase, one of them has been previously depressed, and the other potentiated. By looking at Figure 4, one can see that, despite being delayed in the second phase, the weights undergo the same trajectories for the normal and the mutant mice. The speed up in the normal mouse results from the adaptation of D-serine levels that, raising up, lowers that threshold of potentiation, allowing the previously depressed weight to undergo a fast LTP. However, in the first phase of learning, the weights start from an intermediate value, and threshold adaptation does not have a significant impact.

Finally, in Table 1 is stated that $\Delta w_k = 0$ if the other weight change is non-zero. It is not clear why this should be - it does not seem to be reflected in e.g. Eq. 7a.

We added a modified explanation to the caption of the Table: “We note that, since $\Delta \vec{w}$ is proportional to the one-hot input vectors, only one of the weights changes at each transition”

Reviewers' comments:

Reviewer #1 (Remarks to the Author):

The revised version is improved in its presentation. We now see how the reward-based rule is related to the BCM-like plasticity rule. There are still a few issues that I think should be resolved.

In the section explaining the learning rule:

- Two scaling constants are defined in equations 2 and 4: a and b . The text states that these have "appropriate dimensions". I think these dimensions need to be explicitly stated.
- After equation 5 we are told that ab is set to 1. Why is that? What were the original choices for a and b and if in the end only their product matters, why not introduce just one constant?

The model for the reward dependence:

- We are given the weight change rule in eqs 6-7 but there is no description of the decision variable or how it is computed except in the figure caption (and later in the Methods). This feels incomplete. In fact, if you do not explain here that y is the probability to switch, using a $-R$ term in 7 does not make sense.
- This brings up a question (perhaps naïve) – why not model the probability to stay in a given state?
- On page 6 the authors discuss that avoidance is made up of two factors: switch away from punished state and not going back to it. They correctly state that they are only modelling one of these factors. But the probability to state in a given state would capture both?
- I do not see the discussion about the eligibility trace after equation 7 as warranted or useful here. This is not used in any way in the model or the rest of the paper. This should be moved to the discussion, otherwise it is rather distracting.

I am not completely satisfied with the answers given in the rebuttal for some of the parameter choices:

- The explanation of why the unpunished state is positively rewarded reads rather hand wavy. Perhaps it is enough to say that this is needed technically because if it was not reinforced the task could not be simulated: if the unpunished state R is zero, after the switch the corresponding weight would not decrease.
- Convergence of the weight depends on the initial values being in the basin of attraction of the stable points. The authors state that they picked (0.02, 0.02) and this seems to be true for all the simulations. Why not pick them randomly within appropriate limits? Or would that not really make any difference?
- The authors say that there is no issue with non-convergence due to the saddle-point is not important because its y -coordinate is 0 and this can be handled by clamping the weights to be above 0. Why not do that?
- Looking at figure 6: what is going on with the two unstable points around (0, 0.6) for the Phase 1 and (0.6,0) for Phase 2? Looking at the vector field, one of them is still a saddle? What is the other one?

Going back to the performance curves in figure 4 and 5. I understand where the shoulder comes from, but is this at all something seen in the data? If that is true, it seems as if the mutant mouse would not extinguish the pre-learned behavior gradually, but with a delayed rapid switch away. Speculatively, this is kind of behavior that has been argued to be based on some sort of rule learning (a switch after a rule has been learned and no switch before it). However, here there is no explicit rule representation – just a model free RL rule. This may be an interesting speculation – that the 3-factor BCM rule gives the appearance of rule-based behavior, without any sort of a model-based RL behind it.

Dependence on the reward magnitudes. Since it is clear that in the mutant case one can change the R values so as to have the delay in reversal learning disappear, the most useful thing to do is to compare the performance of the control and mutant "mice" for the same values of the R 's. For example – is the delay still significant if you take $R_1 = -1.5$ for both the control and mutant? Etc? I think this sort of analysis would really strengthen the results of the model. Otherwise it feels like the model requires judicious and ad hoc choices of the key parameters in order to work.

I think the authors should carefully edit the revised version – a lot of the language reads very awkward with overly long complex sentence in passive voice that make reading the paper a bit of a chore. I would suggest going through the text and shortening the sentences as well as putting the wording in active voice as much as possible.

Reviewer #2 (Remarks to the Author):

While the revision has addressed some of my concerns, I remain unconvinced by the paper.

1. While the problem with the serine concentration has now been resolved, eager to map the data onto the BCM model, the firing rate are all relative to y_0 .

The problem is that y_0 is 10 Hz, which is a substantial rate for in vivo firing. It is unclear what happens to rates $< y_0$, which one would definitely expect to occur in vivo.

2. Fig 1d uses the -relative- spike amplitude (and indeed ref 28) shows that overall the absolute amplitude decreases with stim frequency.

It is unclear that plasticity would care about the relative change in amplitude rather than the absolute one.

3. REgarding the data of Fig2, an alternative explanation is that the mutant animals forget more quickly (there is a trend in the data for day3). Moreover, the inset shows that the mutant animals actually learn -faster- than control, because they start at a higher error rate. It remains unclear how this would map onto the model.n

Reviewer #3 (Remarks to the Author):

The authors have done a good job at addressing my previous concerns. I think the model is now clear and well justified.

I have just one minor additional comment:

Eq. 4: "According to the fact that a higher level of D-serine facilitates LTP, we choose the threshold function to be inversely proportional to the D-serine concentration:"

This equation is not inversely proportional to d : (i.e. proportional to $1/d$), rather it is a linearly decreasing function of d .

Dear reviewers,

Thank you for the helpful comments and the opportunity to resubmit a revised version of our manuscript. Below, we address all comments of the referees point-by-point. The referee comments are in **bold font** while our answers are in regular font. The manuscript parts that have been adapted in response to referee feedback are highlighted using the blue font.

Reviewer #1:

The revised version is improved in its presentation. We now see how the reward-based rule is related to the BCM-like plasticity rule. There are still a few issues that I think should be resolved.

In the section explaining the learning rule:

1. **Two scaling constants are defined in equations 2 and 4: a and b . The text states that these have “appropriate dimensions”. I think these dimensions need to be explicitly stated.**

We added an explicit dimensional analysis for both quantities:

Based on this, we choose the simple mathematical form:

$$D(y) = D_0 - a(y - y_0)^2 \quad (2)$$

where D_0 is the maximum level of D-serine concentration, occurring when $y = y_0$, and a is a constant with the appropriate dimensions ($[a] = [D] \cdot [y]^{-2} = [N] \cdot [L]^{-2} \cdot [T]^2$).

And later:

According to the fact that a higher level of D-serine facilitates long-term potentiation (LTP), we choose the threshold function to be linearly decreasing with respect to the D-serine concentration:

$$\theta(d) = b(D_0 - d) \quad (4)$$

where b is a constant with the appropriate dimensions ($[b] = [z] \cdot [d]^{-1} = [T]^{-1} \cdot [N]^{-1} \cdot [L]^2$).

2. **After equation 5 we are told that ab is set to 1. Why is that? What were the original choices for a and b and if in the end only their product matters, why not introduce just one constant?**

Thank you for this interesting suggestion. The reason we defined two distinct constants, a and b , rather than one is to obtain a coherent dimensional analysis and biological derivation in two distinct equations (Eq.2 and 4). The precise values of the constants a and b can be arbitrarily chosen without affecting the qualitative predictions (more formally, we choose arbitrary units of measurement, so that the value of the product is 1). To clarify why we use two rather than one parameter in the model derivation we have added the following sentences after Eq. 5:

To minimize the number of free parameters and be as consistent as possible with respect to previous parameter settings in the BCM model we set $ab = 1$ for all our subsequent model simulations. However, during the model formulation stage, we thought it could be helpful to think of a as a proportionality factor between firing rate and D-serine amount and of b as a proportionality factor mediating the transformation between threshold and D-serine, since both could be altered individually in future studies or modified experimentally.

The model for the reward dependence:

3. **We are given the weight change rule in equations 6-7 but there is no description of the decision variable or how it is computed except in the figure caption (and later in the Methods). This feels incomplete. In fact, if you do not explain here that y is the probability to switch, using a $-R$ term in 7 does not make sense.**

Thank you for the suggestion to clarify this definition. We added a paragraph describing the decision variable and how it is computed before giving the weight change rule, so that now it is not necessary to go to Methods or the figure caption to get such information. The corresponding paragraph reads as follows.

The decision process of the mouse is described by a single action unit (Fig. 3, bottom). Each state S^i is associated with a one-hot encoding vector \vec{x}^i serving as input for the action unit. The agent-mouse decides to switch state with a probability directly proportional to the activation $y(\vec{x}^i)$ of the action unit.

4. **This brings up a question (perhaps naïve) – why not model the probability to stay in a given state?**

Every iteration of the process requires the mouse to stochastically switch state (with probability p) or stay (with probability q). Since those are the only two possibilities, one has $q = 1 - p$ so that modeling one or the other leads to the same behavior. We chose to model the probability of switching just because it seemed easier to interpret conceptually (as a reaction to a negative stimulus).

Let us mention that another possibility could have been to model the probability of the mouse going into state 1 or state 2, which is maybe closer to a reinforcement learning scenario in which the agent decides where to go, but in this model that would have led to trivial behavior in which the mouse always goes in the rewarding state (irrespective of the state it is in), thus practically ignoring the state. For this reason, we decided to model the task as “switch” or “stay”.

5. **On page 6 the authors discuss that avoidance is made up of two factors: switch away from punished state and not going back to it. They correctly state that they are only modeling one of these factors. But the probability to stay in a given state would capture both?**

We thank the reviewer for commenting on this interesting conceptual aspect. Since the model only describes synaptic plasticity, it captures only the “not going back to the punished state” behavior. This can be seen by observing the stationary values of the weights in Fig. 4, bottom graph. The weight representing the probability of switching away from the safe state converges close to zero, while the probability of switching away from the punishing state converges close to 0.5. While the reasoning is intuitive, we found it hard to capture it in words without sounding too convoluted. We simplified the corresponding text while highlighting what we think are crucial aspects. It now reads as follows:

The weight w_2 , which represents the probability of switching state from S^2 to S^1 , converges to a higher value, but lower than 0.5. This means that the probability of leaving the punishing state is not close to 1, as might be intuitively expected from a probabilistic agent who learned to avoid this situation. However, this property can be justified and interpreted as proof of the coherence of the model. The *avoidance* behavior can be conceptually divided into two sub-behaviors: *going away from* the aversive situation and *not going back to* it. Such a distinction is important because the two behaviors are unlikely to arise from the same neurobiological processes. For instance, the *not going back to* behavior requires memory of past experiences, while the *going away from* does not since the aversive stimulus is present and can drive the behavior directly. Coherently, because we are only modeling one mechanism (synaptic plasticity), the model captures the *not going back to* behavior ($P(\text{stay}|\text{no punishment}) \approx 1$) but not the *going away from* ($P(\text{change}|\text{punishment}) \approx 0.5$).

6. **I do not see the discussion about the eligibility trace after equation 7 as warranted or useful here. This is not used in any way in the model or the rest of the paper. This should be moved to the discussion, otherwise it is rather distracting.**

Following the suggestion of the referee we removed it. Thanks for pointing it out.

I am not completely satisfied with the answers given in the rebuttal for some of the parameter choices:

7. **The explanation of why the unpunished state is positively rewarded reads rather hand wavy. Perhaps it is enough to say that this is needed technically because if it was not reinforced the task could not be simulated: if the unpunished state R is zero, after the switch the corresponding weight would not decrease.**

Thank you for pointing out that we should stress more that the reason why we introduced a positive reward is merely technical. We now write:

Let us mention that the need for having a positive signal is due to the form of the R-BCM equation (7): if the value of R for the unpunished state is set to zero, the corresponding weight will not be able to increase after the switch from phase one to phase two.

However, we believe that the explanation provided about it (concerning the fact that it is the combination of the two values that gives rise to the avoidant behavior) is still meaningful for the reader, and we decided to keep it.

8. **Convergence of the weight depends on the initial values being in the basin of attraction of the stable points. The authors state that they picked (0.02, 0.02) and this seems to be true for all the simulations. Why not pick them randomly within appropriate limits? Or would that not really make any difference?**

Thank you for this helpful comment. We chose the initial weights to be equal so that at the beginning of Phase 1 the mouse does not have any preference for the states. If this is not the case, Phase 1 could correspond both to direct learning or reversal learning (when one of the weights is initially higher than the other). Indeed, the starting value of the weights, representing the information previously acquired by the mouse, is the only difference between Phase 1 (direct learning) and Phase 2 (reversal learning). For this reason, we avoid sampling random initial weights. However, as long as the weights are within the basin of attraction, this will not alter the results we present. We plot some examples in the figure below (only for Phase 1, since in Phase 2 the starting weights do not depend on the initial values).

9. **The authors say that there is no issue with non-convergence due to the saddle-point is not important because its y -coordinate is 0 and this can be handled by clamping the weights to be above 0. Why not do that?**

We do not "hard-code" the weight clamping to zero because the dynamics cannot generate negative weights, as long as the initial weights are non-negative.

10. **Looking at figure 6: what is going on with the two unstable points around (0, 0.6) for the Phase**

1 and (0.6, 0) for Phase 2? Looking at the vector field, one of them is still a saddle? What is the other one?

Yes, one of them is still a saddle point, while the other is an unstable node. In practice, the fixed point lying on the axis $y = 0$ (or $x = 0$) is always a saddle-node (before and after the switching), while the fixed point close to it is a stable (or unstable) node in Phase 1, and it becomes an unstable (or stable) node in Phase 2. We plot below a zoom of the vector field in Phase 1 and Phase 2, for one of the pair of fixed points.

11. **Going back to the performance curves in figure 4 and 5. I understand where the shoulder comes from, but is this at all something seen in the data? If that is true, it seems as if the mutant mouse would not extinguish the pre-learned behavior gradually, but with a delayed rapid switch away. Speculatively, this is kind of behavior that has been argued to be based on some sort of rule learning (a switch after a rule has been learned and no switch before it). However, here there is no explicit rule representation – just a model free RL rule. This may be an interesting speculation – that the 3-factor BCM rule gives the appearance of rule-based behavior, without any sort of a model-based RL behind it.**

The behavior of mice seems to follow a devaluation in reversal learning congruent with model-free agents (i.e., there are several revisits to the airpuff area after switching its position). However, it is hard with an unconditioned set-up to discern how states are mapped. Moreover, the absence of task rules impedes us to come to a conclusion regarding the mechanism involved. Both Figures 4 and 5 show a slow reversal of each state.

The hypothesis is very interesting, yet, our algorithmic implementation follows normative model-free learning and future work building on our results could combine biologically plausible rules combining local learning with global reinforcers.

12. **Dependence on the reward magnitudes. Since it is clear that in the mutant case one can change the R values so as to have the delay in reversal learning disappear, the most useful thing to do is to compare the performance of the control and mutant “mice” for the same values of the R ’s. For example – is the delay still significant if you take $R_1 = -1.5$ for both the control and mutant? Etc? I think this sort of analysis would really strengthen the results of the model. Otherwise it feels like the model requires judicious and ad hoc choices of the key parameters in order to work.**

The comparison of the performance of the control and mutant “mice” for the same values of the R ’s is already present in the graph of Fig. 5, since we simulate with $R = -1, -1.2, -1.5$ for the mutant, and $R = -1$ for control. We modified the caption of the figure to emphasize it. In the case of $R = -1.5$ for both control and mutant, as suggested by the reviewer, the control mice still learn faster than the mutant mice. We plot the graph below. The speed difference is not as significant as in the case $R = -1$. We think this behavior is reasonable since it would seem unnatural for the learning speed to increase indefinitely as the aversiveness of the stimulus increases. However, since the aversiveness of the stimulus was not varied during the original experiment of ref 28, it is hard to evaluate the model on this.

13. **I think the authors should carefully edit the revised version – a lot of the language reads very awkward with overly long complex sentence in passive voice that make reading the paper a bit of**

a chore. I would suggest going through the text and shortening the sentences as well as putting the wording in active voice as much as possible.

Thank you for pointing this out. We carefully read the manuscript and improved its readability by shortening sentences and removing passive voices. In particular, we have majorly edited the Introduction, improving the language and the clarity. We have also included further proofreading by group members to optimize readability.

Reviewer #2:

While the revision has addressed some of my concerns, I remain unconvinced by the paper.

1. While the problem with the serine concentration has now been resolved, eager to map the data onto the BCM model, the firing rate are all relative to y_0 . The problem is that y_0 is 10 Hz, which is a substantial rate for in vivo firing. It is unclear what happens to rates $< y_0$, which one would definitely expect to occur in vivo.

Thank you for these insightful comments about the firing rate regime considered by our model and the interpretation of the experimental data on which our model is based. To make sure that our reply and the strategy we follow in the design of the model, as well as the interpretation of the behavioral measurements by Bohmbach et al is sound, we have consulted a senior expert in the field of astrocytic and neural in vivo dynamics, Prof. Valentin Naegerl from the Universite de Bordeaux. We asked specifically Prof. Valentin Naegerl because he has unique experience working with computational models and leading experiment-theory collaborations that drove the development of the tripartite synapse models. His expertise was thus unique in helping us judge the merits and caveats of our model, and if necessary adapt. He confirmed that our model provides a novel and unique angle that allows for building quantitative hypotheses for the computational role of an astrocytic player and would be of interest to a broad range of physiologists, modelers, and behavioral scientists. Since such models are currently sparse we believe that our work closes an important gap.

Let us comment specifically on the firing rate regime where our model operates and why we believe it accurately captures the biology. The firing rate in vivo has been reported to be highly variable and can depend on the region, the recording technique, the state of the animal, and the type of stimulus among other factors. With the firing rate regime in our model, we aimed to target the in vivo firing rate reported specifically for hippocampal pyramidal CA1 neurons in animals that are actively behaving and learning. Experimental studies indicate this firing range to be 5 – 20Hz (Wiener, Paul, and Eichenbaum 1989; Czurkó et al. 1999; Hirase et al. 1999). For example, when searching the curated online collection (<https://bionumbers.hms.harvard.edu/>) for “CA1 pyramidal cells firing rate in vivo” referred to a publication stating “Although single-unit recordings in vivo indicate that CA1 pyramidal cells fire at approximately 1 Hz at rest, during activity (such as running in a wheel)

the firing rate increases to 5-20 Hz and may be sustained throughout the duration of the activity” indicating that the firing rate regime we chose for our model is in line with in previous experimental reports for CA1 pyramidal cells. Our model describes the firing rate of CA1 pyramidal neurons during exploration and learning, thus we chose 10Hz as a reference point for our model because it is both consistent with the experimental ”mean” in vivo activity and is consistent with the peak experimentally reported D-serine activity in our previous works including Bohmbach et al. To stress that our model operates we have added the following statement to the model introduction:

We define the integration of the input vector with the synaptic strengths to track the neural activity with respect to the reference value ν_0 , thus $y = \nu - \nu_0 = \vec{w} \cdot \vec{x}$. The operating regime of our model will be above ν_0 , $y > 0$ because this is best supported by the available experimental data. Let us mention that firing rates below this regime could be described by other plasticity mechanisms that are outside of our model framework.

Furthermore, let us note that the value $\nu_0 = 10Hz$ is just a rough estimate of the peak of the bell-shaped function measured in ref 28. Since the function was sampled only at four distinct firing rates (4, 10, 20, 40Hz, see Fig. 1d) and the maximum value was found for 10Hz, we can only conclude that the peak lies in the range 4 – 20Hz. We stress that our model outcome and the proposed mechanism do not rely on a specific value of ν_0 therefore, aiming for a reference point that is a round number, consistent with the in vivo activity regime, we chose 10Hz.

2. **Fig 1d uses the -relative- spike amplitude (and indeed ref 28) shows that overall the absolute amplitude decreases with stim frequency. It is unclear that plasticity would care about the relative change in amplitude rather than the absolute one.**

In ref 28 the amplitude of dendritic spike is used only as a proxy for measuring the extracellular concentration of D-serine. This is possible because a correlation between the amplitude and the levels of D-serine was reported previously. In our work, we relate the changes in D-serine concentration to changes in synaptic plasticity, according to ref 27. Thus, the relationship between dendritic spikes amplitude and synaptic plasticity is not part of the model and our model predictions do not depend on it.

3. **Regarding the data of Fig2, an alternative explanation is that the mutant animals forget more quickly (there is a trend in the data for day 3). Moreover, the inset shows that the mutant animals actually learn -faster- than control, because they start at a higher error rate. It remains unclear how this would map onto the model.**

The alternative explanation that the mutant animals forget more quickly is not supported by statistical tests, since there is no trend on day 3. The impression of a trend might be due to the skewed distribution of the data represented by the IQR plot, but the medians (the straight line) are at the same height. We added the results of the statistical tests in the caption of the figure. On the contrary, the trend on day 4 is highly significant, clearly indicating that mutant mice exhibit a slow-down in the reverse phase of learning. To be convinced of that, refer to ref 28, in particular to Supp. Fig 8L. This graph quantifies time on day 4 and shows clearly that the mutant mice require more time to learn.

Because it regards the inset, it does not show that the mutant animals start with a higher error rate. The ‘error rate’ in the first five minutes of the mutants is higher because they do not learn the new position within those five minutes. However, the wild types do. The mutants need more time to catch up (not until the second five-minute interval); thus, they are slower. To avoid that readers might misunderstand the graph (extracted from ref 28), we now provide more context, by including a detailed explanation in the caption of the figure.

The new caption reads as follows:

(b) Mutant mice showed slower learning with respect to control mice, only during reverse learning (Day 4). On each day, the mouse is free to explore the maze for a single session of 10 minutes, and the number of visits to the puff position is recorded. The box plots represent the statistics from 15 (control) and 17 (mutant) animals. There is no significant difference between the groups on Day 1, 2, and 3 (Day 1: 6.59 ± 0.73 vs. 6.13 ± 0.62 , $p = 0.62$; Day 2: 1.41 ± 0.21 vs. 1.40 ± 0.34 , $p = 0.73$; Day 3: 6.00 ± 1.50 vs. $7.271.45$, $p = 0.45$; two-sided Mann–Whitney U-test), while there is a very significant difference on Day 4 (Day 4: $3.650.50$ vs. $5.930.61$, $p = 0.0066$, two-sided Student’s t test). The inset axis shows the same data for Day 4, binned into two 5-minutes intervals. Each point is computed as the average over all the animals, and the error bar is the standard error. In the first 5 minutes the mutant mice visited the puff position significantly more than the control mice, showing a learning deficit. In the last five minutes, the difference is no more observed. Figures adapted from [28].

Reviewer #3:

The authors have done a good job at addressing my previous concerns. I think the model is now clear and well justified.

1. I have just one minor additional comment: Eq. 4: *According to the fact that a higher level of D-serine facilitates LTP, we choose the threshold function to be inversely proportional to the D-serine concentration: This equation is not inversely proportional to d : (i.e. proportional to $\frac{1}{d}$), rather it is a linearly decreasing function of d .*

Thank you for pointing out this mistake. We corrected it and we report the new text below.

According to the fact that a higher level of D-serine facilitates LTP, we choose the threshold function to be linearly decreasing with respect to the D-serine concentration:

$$\theta(d) = b(D_0 - d) \tag{1}$$

References

- Czurkó, András et al. (1999). “Sustained activation of hippocampal pyramidal cells by ‘space clamping’ in a running wheel”. In: *European Journal of Neuroscience* 11.1, pp. 344–352.
- Hirase, Hajime et al. (1999). “Firing rate and theta-phase coding by hippocampal pyramidal neurons during ‘space clamping’”. In: *European Journal of Neuroscience* 11.12, pp. 4373–4380.
- Wiener, Sidney I, CA Paul, and H Eichenbaum (1989). “Spatial and behavioral correlates of hippocampal neuronal activity”. In: *Journal of Neuroscience* 9.8, pp. 2737–2763.

REVIEWERS' COMMENTS:

Reviewer #1 (Remarks to the Author):

I would like to thank the authors for the efforts and care they took to answer my questions. At this point, I am satisfied that all the major issues I raised have been clarified and I believe the manuscript can proceed.